# RefAM: Attention Magnets for Zero-Shot Referral Segmentation

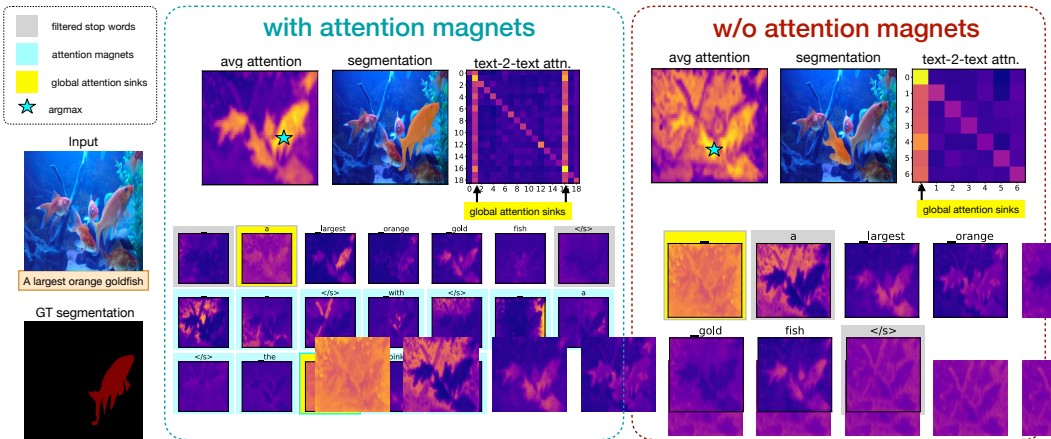

Figure 1: **Global Attention Sinks (GAS) in DiT.** We highlight tokens (here, tokens #1 and #16) that act as GAS in late layers. These tokens allocate disproportionately high and nearly uniform attention across all text and image tokens simultaneously. GAS are absent in early layers, emerge consistently in deeper blocks, and serve as indicators of semantic structure. While uninformative themselves, they can suppress useful signals when they occur on meaningful tokens.

## ABSTRACT

Most existing approaches to referring segmentation achieve strong performance only through fine-tuning or by composing multiple pre-trained models, often at the cost of additional training and architectural modifications. Meanwhile, large-scale generative diffusion models encode rich semantic information, making them attractive as general-purpose feature extractors. In this work, we introduce a new method that directly exploits features, attention scores, from diffusion transformers for downstream tasks, requiring neither architectural modifications nor additional training. To systematically evaluate these features, we extend benchmarks with vision–language grounding tasks spanning both images and videos. Our key insight is that stop words act as attention magnets: they accumulate surplus attention and can be filtered to reduce noise. Moreover, we identify global attention sinks (GAS) emerging in deeper layers and show that they can be safely suppressed or redirected onto auxiliary tokens, leading to sharper and more accurate grounding maps. We further propose an attention redistribution strategy, where appended stop words partition background activations into smaller clusters, yielding sharper and more localized heatmaps. Building on these findings, we develop REFAM, a simple training-free grounding framework that combines cross-attention maps, GAS handling, and redistribution. Across zero-shot referring image and video segmentation benchmarks, our approach consistently outperforms prior methods, establishing a new state of the art without fine-tuning or additional components.

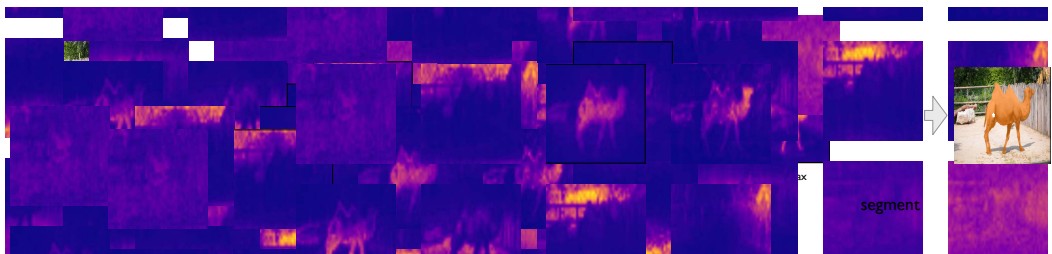

Figure 2: **Pipeline overview.** We first extract cross-attention maps for the referring expression with attention magnets. Next, we filter out stop words and attention magnets, aggregate the remaining maps, identify the argmax location, and apply SAM to generate the final segmentation mask.

# 1 INTRODUCTION

Diffusion transformers (DiTs) have rapidly advanced generative modeling and, more recently, been adopted as powerful feature extractors for downstream vision–language tasks such referring object segmentation (Ni et al., 2023). Their cross-attention maps encode rich spatial and semantic information without task-specific training, making them attractive for training-free and zero-shot applications. However, attention in transformers is also known to exhibit emergent behaviors that are not always semantically meaningful. In large language models, for instance, certain tokens, often first tokens, attract disproportionately high attention while carrying little to no semantic content, a phenomenon referred to as attention sinks or massive activations (Xiao et al., 2024b; Yona et al., 2025; Jin et al., 2025; Sun et al., 2024a; Barbero et al., 2025).

In this paper, we extend this observation to generative diffusion transformers and show that they exhibit similar attention sink behaviors when applied to vision–language grounding tasks. Specifically, we uncover language–vision attention sinks, where stop words emerge as high-attention tokens despite lacking semantic value. We find two distinct patterns. First, a small set of stop words consistently act as global attention sinks (GASs) in the later layers of DiTs: they attend almost uniformly across text and image tokens, and filtering their channels does not harm downstream performance. Second, other stop words behave as local background attractors, drawing attention toward irrelevant regions. Surprisingly, appending additional stop words introduces more such attractors, which redistributes background attention and yields cleaner heatmaps. We further show that replacing stop words with random vectors also improves results, but real stop words are consistently more effective, likely due to their repeated presence during pretraining.

These findings suggest that stop words can serve as a simple yet effective tool for attention redistribution. Building on this, we propose REFAM, a training-free grounding method, that augments referring expressions with stop words, filters their attention maps, and aggregates the remaining cross-attention for grounding. This approach requires no modifications to the diffusion model, no additional supervision, and generalizes to both image and video tasks.

In summary, our contributions are threefold:

- We identify and analyze global attention sinks (GASs) with respect to both language and visual tokens in DiTs, linking their emergence to semantic structure and showing that they carry no useful signal for grounding.

- We introduce a stop-word based attention redistribution strategy, where added stop words act as magnets that absorb surplus attention and enable cleaner cross-attention maps.

- We achieve state-of-the-art zero-shot referring segmentation on both image and video benchmarks using REFAM, features from diffusion transformers, outperforming prior training-free methods without fine-tuning or auxiliary components.

We will make all our code public to make our results reproducible for the broader community.

## 2    RELATED WORK

**High-Norm Tokens Across Transformer Architectures.**    Recent research has identified tokens exhibiting high-norm activations across various domains, including language models (Xiao et al., 2024b; Yona et al., 2025; Jin et al., 2025; Sun et al., 2024a; Barbero et al., 2025), vision models (Kang et al., 2025; Darcet et al., 2024b; Jiang et al., 2025; Wang et al., 2024a), and vision-language models (An et al., 2025; Woo et al., 2024). In language models, these tokens are referred to as attention sinks (Xiao et al., 2024b; Yona et al., 2025; Barbero et al., 2025) or massive activations (Jin et al., 2025; Sun et al., 2024a). In vision models, similar phenomena are termed registers (Darcet et al., 2024b; Jiang et al., 2025), visual attention sinks (Kang et al., 2025), or defective path tokens (Wang et al., 2024a). In vision-language models, this phenomenon has been described as attention deficiency (An et al., 2025) or blind tokens (Woo et al., 2024), reporting individual visual tokens that consistently receive disproportionately high attention. These studies consistently show that a small fraction of tokens absorb disproportionately high attention, often without semantic relevance. In our work, we identify global attention sinks (GAS), tokens that span both language and visual streams under the same query and systematically suppress useful signals in both modalities, simultaneously.

A common explanation attributes this behavior to the softmax normalization constraint: attention weights must sum to one, even when the query lacks a strong contextual match (Xiao et al., 2024b). In such cases, attention is distributed toward tokens that act as "sinks". Other studies offer complementary views: Jin et al.(Jin et al., 2025) point to positional encodings; Sun et al.(Sun et al., 2024a) implicate learned bias terms; and Wang et al. (Wang et al., 2024a) connect the effect to the power method, where repeated matrix multiplications amplify dominant directions in feature space. While many approaches aim to mitigate this behavior, we instead draw on it: our method introduces attention magnets, tokens, *e.g.,* stop words, that deliberately absorb surplus attention to help redistribute focus more effectively in generative models. Unlike Darcet et al. (Darcet et al., 2024a), who require re-training the model with learnable "register" tokens to accumulate global information, our approach is training-free. We identify that multimodal attention sinks emerge naturally in pre-trained DiTs and that existing stop words effectively fulfill this role. Furthermore, while registers are designed to store information, our attention magnets function as garbage collectors, filtering out noise to sharpen the grounding signal.

**Referring object segmentation.**    Referring object segmentation aims to localize a region in an image or video based on a natural language expression. We address both static and temporal settings. (1) *Referring Image Object Segmentation (RIOS).* Traditional methods are supervised (Kazemzadeh et al., 2014; Ding et al., 2020; Feng et al., 2021; Li et al., 2018), but recent zero-shot approaches leverage pre-trained models. Global-Local (Yu et al., 2023) extracts CLIP-based features from mask proposals, and Ref-Diff (Ni et al., 2023) uses diffusion priors. HybridGL (Liu & Li, 2025) fuses global-local context with spatial cues. These methods first generate multiple object proposals and then score masks by their similarity to text embeddings. (2) *Referring Video Object Segmentation (RVOS).* Temporal methods like LoSh (Yuan et al., 2024) and WRVOS (Zhao et al., 2023) require supervision. In contrast, recent zero-shot methods such AL-Ref-SAM (Ren et al., 2024) use pre-trained image grounding GroundedSAM (Ren et al., 2024) and SAM2 (Ravi et al., 2025) models for frame-wise grounding with minimal adaptation. Unlike prior approaches, our method unifies both tasks under a single framework using diffusion-derived features, operating in a training-free, zero-shot setting.

## 3    SEMANTIC FEATURES FROM DIFFUSION TRANSFORMERS

We describe how semantic features are extracted from rectified-flow diffusion transformers (DiTs) and how stop words, acting as *attention magnets*, enable robust referral segmentation. Our method consists of three components: (i) extracting cross-attention maps from DiTs, (ii) identifying and filtering attention sinks, and (iii) redistributing surplus attention through additional tokens (Figure 2).

### 3.1    FEATURE EXTRACTION FROM DITS

Rectified flow models (Labs, 2024; Team, 2024) combine an image-to-latent encoder–decoder with a DiT (Peebles & Xie, 2023) backbone. The encoder compresses inputs into a latent space, while the DiT performs denoising through a sequence of transformer blocks. Architectures may interleave

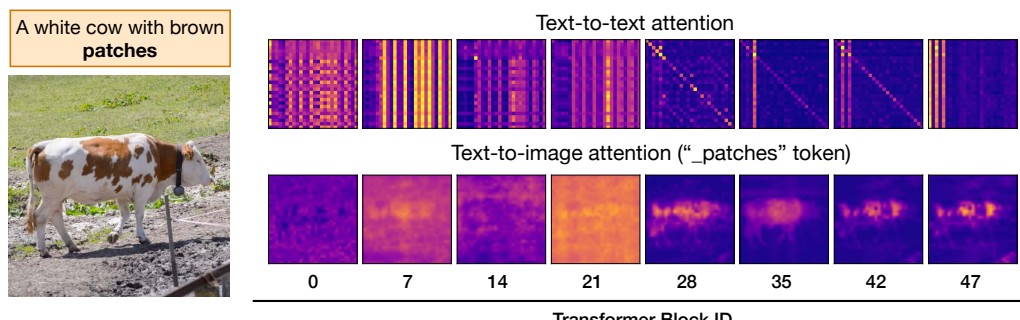

Figure 3: **Emergence of semantic information in DiT.** Top: text-to-text attention across layers. Early layers (0–19) are diffuse and uniform, while middle and late layers (20–47) develop block-diagonal structure, indicating meaningful linguistic grouping. Bottom: text-to-image attention for the "_patches" token. Early layers spread attention broadly over the scene, whereas middle layers begin to localize, and late layers sharpen around the target object. These dynamics illustrate how semantic alignment emerges progressively with depth.

double-stream blocks, which process text and visual tokens separately before merging in attention, and single-stream blocks, which operate on concatenated tokens with shared weights.

Given a clean latent $X_0$, the rectified flow forward process perturbs it as

$$X_t = (1 - \sigma)X_0 + \sigma\epsilon, \quad \epsilon \sim \mathcal{N}(0, I).$$

While the DiT is trained to predict the noise $\epsilon$, its intermediate activations capture rich semantic information. In particular, cross-attention maps between text and image tokens provide spatial grounding signals. For the denoising process itself, the model uses either the source prompt (if available) or an empty prompt. In parallel, we collect features from a separate text branch that encodes the referring expression, similarly as in (Helbling et al., 2025). This branch is used exclusively for feature extraction and has no effect on the denoising trajectory. Unlike prior work that primarily relies on U-Net features (Tang et al., 2023; Zhang et al., 2023), we exploit these transformer attention maps directly, which we find more effective for referring segmentation.

### 3.2 REFERRAL OBJECT SEGMENTATION

The goal of referring object segmentation is to localize a target region in an image or video given a natural language expression. Formally, for an input $(I, e)$, where $I$ is an image or a video frame and $e$ is a referring expression, the task is to predict a segmentation mask $m$ that highlights the region described by $e$.

**Cross-attention features.** Following Concept Attention (CA) (Helbling et al., 2025), we use cross-attention maps as grounding signals. Unlike CA, which assumes access to *all relevant concepts* in the image, our setting is more realistic: only the referring expression $e$ is provided. For each token $t_k \in e$, we extract cross-attention maps $M^{(k)}$ from multiple layers and heads of the DiT, then aggregate them into a consolidated heatmap $H_e$. The referred location is obtained as

$$p_{\text{ref}} = \arg\max H_e.$$

**Stop-word augmentation and filtering.** During attention computation, stop words frequently attract disproportionately high attention (see Fig. 1), which degrades localization precision. We turn this phenomenon into an advantage through a two-step procedure. *First*, we augment the expression $e$ by appending additional stop words (e.g., ".", "a", "with"), producing an expanded expression $\hat{e}$. *Second*, we filter out attention maps corresponding to stop words when aggregating token-level maps. Formally,

$$H_e = \text{mean}\{M^{(k)} \mid t_k \in \hat{e}, \ t_k \notin S_{\text{stop}}\},$$

where $S_{\text{stop}}$ is a predefined set of stop words, extended with tokenizer-specific symbols (".", "," and "_"). Appended stop words act as *attention magnets*, absorbing surplus background activations; discarding them yields sharper, less cluttered heatmaps.

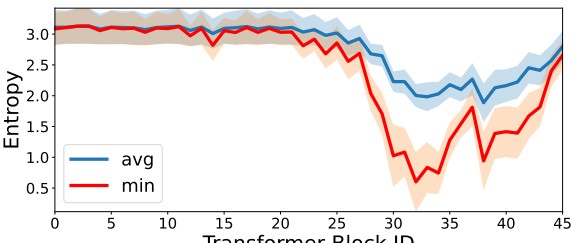

Figure 4: **Entropy across transformer blocks.** Blocks 0-25 contain no specific information.

| Filtered Blocks | $\mathcal{J}\&\mathcal{F}$ | $\mathcal{J}$ | $\mathcal{F}$ |
|---|---|---|---|
| 80 % | 54.3 | 51.2 | 57.5 |
| 70 % | 56.3 | 53.2 | 59.4 |
| 60 % | 57.6 | 54.6 | 60.6 |
| 0 % | 57.6 | 54.5 | 60.6 |

Table 1: **RVOS performance dependence on % of filtered transformer blocks.** Performance starts degrading only after filtering more than 60% of blocks.

**Segmentation.** Our method is model-agnostic and applies to both images (FLUX (Labs, 2024)) and videos (Mochi (Team, 2024)). For images, we convert the attention heatmap into a segmentation mask using a foundation model such as SAM or SAM2 (Kirillov et al., 2023; Ravi et al., 2025). For videos, we extract the query point from the first frame and propagate the segmentation across the sequence with SAM2. In both cases, the pipeline is entirely training-free and operates in a zero-shot setting.

### 3.3 Emergence of Semantic Information in DiT

We next examine how semantic structure arises across transformer blocks in diffusion transformers (DiTs). As shown in Fig. 3, text-to-text and text-to-image attention evolve from diffuse to semantically structured with depth.

**Early layers** *(diffuse attention).* In the initial blocks (0–16), both text and image tokens attend broadly and diffusely. Attention maps are uniform, producing little usable alignment for grounding (Fig. 3, blocks 0-25). In particular, we show in Fig. 4 that 60% of the transformer blocks contain no structured information as the average and minimal entropy of the blocks remains high. Moreover, in Tab. 1 we demonstrate that filtering these blocks does not change the performance.

**Middle layers** *(clustering and alignment).* From mid-level blocks onward, structure begins to emerge: image tokens form clusters corresponding to coarse regions, while text tokens specialize toward different spatial areas. For example, the "␣patches" token in Fig. 3 gradually concentrates on the brown patches on the animal's body. This stage marks the onset of meaningful cross-modal alignment.

**Late layers** *(emergence of GAS).* In later layers, semantic alignment sharpens, but we also consistently observe *Global Attention Sinks (GAS)*. These are tokens, most often stop words, that allocate unusually high and nearly uniform attention across both text and image tokens (Fig. 1). We identify GAS by computing per-token text-to-text activations and marking tokens whose average mass is $10\times$ higher than the mean across all layers and all tokens. In Fig. 1, we visualize layer-averaged text-to-text and text-to-visual attention for each token. On the left, tokens #1 ('␣a') and #16 ('') exhibit uniformly high attention across both textual and visual tokens, characteristic of global attention sinks. Typically, 1–3 GAS tokens appear per sequence.

### 3.4 Interpretation of Global Attention Sinks

Next, we analyze the role of Global Attention Sinks (GAS) and their impact on referral segmentation.

**Uninformative role.** While GAS tokens serve as indicators of emerging semantic structure (see Fig. 3), they do not encode meaningful content. Removing them has no negative effect on performance; when suppressed during inference, their surplus activations are naturally redistributed to non-sink tokens, confirming that their contribution is noise-like rather than semantically useful.

**Indicators of semantic structure.** GAS consistently emerge only after meaningful structure is established in the middle layers. Their appearance therefore marks the onset of semantically organized representations, even if the GAS tokens themselves are uninformative.

**Potentially harmful role.** While the majority of GAS tokens (77%) correspond to stop words, about 10% fall on color tokens and another 10% on other content words. In these cases, GAS behavior can suppress discriminative cues (e.g., color specificity), suggesting untapped headroom if such suppression were prevented.

## 3.5 REDISTRIBUTION STRATEGY WITH ATTENTION MAGNETS

The distribution of semantic information across tokens in later layers raises two challenges for referral segmentation: (i) GAS tokens that suppress meaningful content when they fall on discriminative tokens, and (ii) background activations that contaminate attention maps. We address both through redistribution with *attention magnets*—appended tokens that attract surplus attention and are later filtered out.

**(i) Redistributing GAS.** When GAS fall on stop words, they are harmless. However, when they occur on meaningful tokens such as colors, they erase discriminative distinctions. By appending auxiliary magnets (extra stop words and color words), we redirect uniform attention away from these tokens. Empirically, in ∼89% of cases, color-GAS tokens reassign their mass to the magnets, allowing the original tokens (e.g., "red", "white") to recover specificity.

**(ii) Redistributing background attention.** Even in the absence of GAS, stop words act as local magnets that absorb surplus attention from irrelevant regions such as sky, ground, or background objects. A single or small set of stop words often clusters large areas into one diffuse blob, which still contaminates the averaged heatmap. By appending additional stop words with diverse embeddings, we increase the number of available magnets. This partitions the background into multiple smaller clusters, each absorbed by a different magnet. After filtering these tokens, the residual heatmaps are sharper and contain less clutter, see Fig. 5.

**Practical effect.** The combined mechanism, (i) redirecting global sinks into magnets and (ii) partitioning background noise across multiple attractors, consistently improves grounding. Foreground maps become sharper and more concentrated, while meaningful tokens preserve their semantic roles. Crucially, the approach is entirely training-free: it leverages inductive behavior already learned during pretraining (e.g., frequent exposure to stop words) rather than introducing new parameters. This strategy is grounded in recent NLP findings where specific tokens (e.g., punctuation or start-of-sentence tokens) act as attention sinks to stabilize inference (Xiao et al., 2024a). We observe a parallel phenomenon in multimodal DiTs: stop words naturally attract surplus attention mass. By explicitly appending these "magnets," we provide a designated destination for background noise, preventing it from contaminating semantic tokens.

## 4 RESULTS

We evaluate our proposed method on referring image object segmentation (RIOS), and referring video object segmentation (RVOS). For each task, we compare against state-of-the-art baselines under training-free settings.

**Datasets.** We evaluate our method on the standard benchmarks for referring image and video segmentation tasks. For referring image segmentation (RIOS), we use RefCOCO/+/g (Kazemzadeh et al., 2014), containing referring expressions for objects in COCO images (Lin et al., 2014). For referring video segmentation (RVOS), we use Ref-DAVIS17 (Khoreva et al., 2019), Ref-YouTube-VOS (Seo et al., 2020) and MeViS (Ding et al., 2023), which provides video object masks and expressions for sequences. MeViS is a newly established dataset that is targeted at motion information analysis and its test set consists of 50 videos and 793 annotations. The Ref-YouTube-VOS stands out as the most extensive R-VOS dataset, comprising 202 videos and 834 annotations. Ref-DAVIS17 builds upon DAVIS17 (Khoreva et al., 2019) and contains 30 videos with 244 annotations.

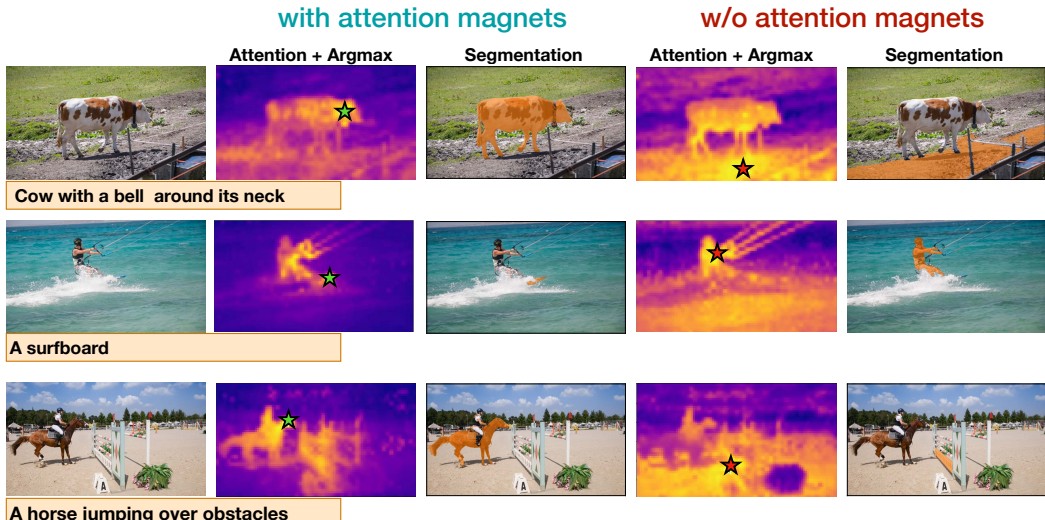

Figure 5: **Influence of attention magnets on RVOS.** Examples demonstrating attention magnets filtering impact.

**Implementation Details.** As attention magnets, we append "_", "with", "to", "and" stop words and "pink" as and auxiliary color that redistributes some of the meaningful GAS tokens from the referral expression to ours attention magnets. We filter out stop words used not only as attention magnets, but also stop words within the referring expressions, and the end-of-sequence (¡/s¿) token. We use the same preprocessing strategy as HybridGL (Liu & Li, 2025) to extract noun phrases (NP) and spatial bias (SB) from the referring expression using spacy package, ensuring a fair comparison. For referring image object segmentation (RIOS), we use the FLUX model (Labs, 2024) and collect features from timestep 750. For referring video object segmentation (RVOS), we use the Mochi model (Team, 2024) and collect features from timestep 990. To produce final attention map, we aggregate attention maps across all transformer blocks if not stated differently. Since the COCO dataset already provides captions with its annotations, we use these directly to guide feature extraction. We use chatGPT4o to generate captions for DAVIS, Ref-YouTube-VOS and MeViS test videos.

## 4.1 SOTA COMPARISON

We evaluate REFAM on referral image object segmentation (RIOS) on RefCOCO, RefCOCO+, and RefCOCOg datasets using oIoU and mIoU metrics in Tab. 2 and on referral video object segmentation (RVOS) Ref-DAVIS17, Ref-YouTube-VOS and MeViS datasets using standard $\mathcal{J}\&\mathcal{F}$ metrics in Tab. 3. Our method significantly outperforms prior training-free approaches on both RIOS and RVOS, establishing new state-of-the-art results across all datasets. Notable RIOS baselines include Ref-Diff (Ni et al., 2023), MaskCLIP (Zhou et al., 2022), Global-Local (Yu et al., 2023), and the recent HybridGL (Liu & Li, 2025), which rely on complex modeling of spatial or relational cues. In contrast, our approach is simple and leverages the semantic structure learned by pretrained generative models. In particular, compared to HybridGL—the strongest prior zero-shot method—REFAM achieves an absolute gain of +2.5 mIoU on RefCOCOg test and +1.8 mIoU on RefCOCO+ testB. On RefCOCO+ testA, REFAM improves mIoU by more than 9 points over Ref-Diff and by over 12 points over Global-Local. Despite relying only on frozen FLUX features and SAM segmentation method, our approach achieves performance competitive with, and in some cases exceeding, methods that incorporate additional task-specific training or fine-tuning. In Tab. 3, REFAM outperforms all prior training-free baselines and narrowing the gap to recent methods such as Grounded-SAM (Kirillov et al., 2023), Grounded-SAM2, and AL-Ref-SAM (Ren et al., 2024), which are pretrained with image grounding datasets. These results demonstrate that carefully leveraging diffusion features, without retraining, is sufficient to close the gap with supervised and weakly supervised methods, while maintaining the simplicity and generality of a fully training-free pipeline.

| Metric | Method | Vision Backbone | Pre-trained Model | RefCOCO val | RefCOCO testA | RefCOCO testB | RefCOCO+ val | RefCOCO+ testA | RefCOCO+ testB | RefCOCOg val | RefCOCOg test |
|---|---|---|---|---|---|---|---|---|---|---|---|
| | *zero-shot methods w/ additional training* | | | | | | | | | | |
| | Pseudo-RIS (Yu et al., 2024) | ViT-B | SAM, CoCa, CLIP | 37.33 | 43.43 | 31.90 | 40.19 | 46.43 | 33.63 | 41.63 | 43.52 |
| | VLM-VG (Wang et al., 2024b) | R101 | COCO*, VLM-VG* | 45.40 | 48.00 | 41.40 | 37.00 | 40.70 | 30.50 | 42.80 | 44.10 |
| | *zero-shot methods w/o additional training* | | | | | | | | | | |
| | Grad-CAM (Selvaraju et al., 2017) | R50 | SAM, CLIP | 23.44 | 23.91 | 21.60 | 26.67 | 27.20 | 24.84 | 23.00 | 23.91 |
| | MaskCLIP (Zhou et al., 2022) | R50 | SAM, CLIP | 20.18 | 20.52 | 21.30 | 22.06 | 22.43 | 24.61 | 23.05 | 23.41 |
| oIoU | Global-Local (Yu et al., 2023) | R50 | FreeSOLO, CLIP | 24.58 | 23.38 | 24.35 | 25.87 | 24.61 | 25.61 | 30.07 | 29.83 |
| | Global-Local (Yu et al., 2023) | R50 | SAM, CLIP | 24.55 | 26.00 | 21.03 | 26.62 | 29.99 | 22.23 | 28.92 | 30.48 |
| | Global-Local (Yu et al., 2023) | ViT-B | SAM, CLIP | 21.71 | 24.48 | 20.51 | 23.70 | 28.12 | 21.86 | 26.57 | 28.21 |
| | Ref-Diff (Ni et al., 2023) | ViT-B | SAM, SD, CLIP | 35.16 | 37.44 | 34.50 | 35.56 | 38.66 | 31.40 | 38.62 | 37.50 |
| | TAS (Suo et al., 2023) | ViT-B | SAM, BLIP2, CLIP | 29.53 | 30.26 | 28.24 | 33.21 | 38.77 | 28.01 | 35.84 | 36.16 |
| | HybridGL (Liu & Li, 2025) | ViT-B | SAM,CLIP | 41.81 | 44.52 | 38.50 | 35.74 | 41.43 | 30.90 | 42.47 | 42.97 |
| | REFAM (ours) | DiT | SAM, FLUX | **46.91** | **52.30** | **43.88** | **38.57** | **42.66** | **34.90** | **45.53** | **44.45** |
| | *weakly-supervised methods* | | | | | | | | | | |
| | CLRL (Lee et al., 2023) | ViT-B | - | 31.06 | 32.30 | 30.11 | 31.28 | 32.11 | 30.13 | 32.88 | - |
| | PPT (Dai & Yang, 2024) | ViT-B | SAM | 46.76 | 45.33 | 46.28 | 45.34 | 45.84 | 44.77 | 42.97 | - |
| | *zero-shot methods w/ additional training* | | | | | | | | | | |
| | Pseudo-RIS (Yu et al., 2024) | ViT-B | SAM, CoCa, CLIP | 41.05 | 48.19 | 33.48 | 44.33 | 51.42 | 35.08 | 45.99 | 46.67 |
| | VLM-VG (Wang et al., 2024b) | R101 | COCO*, VLM-VG* | 49.90 | 53.10 | 46.70 | 42.70 | 47.30 | 36.20 | 48.00 | 48.50 |
| | *zero-shot methods w/o additional training* | | | | | | | | | | |
| mIoU | Grad-CAM (Selvaraju et al., 2017) | R50 | SAM, CLIP | 30.22 | 31.90 | 27.17 | 33.96 | 25.66 | 32.29 | 33.05 | 32.50 |
| | MaskCLIP (Zhou et al., 2022) | R50 | SAM, CLIP | 25.62 | 26.66 | 25.17 | 27.49 | 28.49 | 30.47 | 30.13 | 30.15 |
| | Global-Local (Yu et al., 2023) | R50 | FreeSOLO, CLIP | 26.70 | 24.99 | 26.48 | 28.22 | 26.54 | 27.86 | 33.02 | 33.12 |
| | Global-Local (Yu et al., 2023) | R50 | SAM, CLIP | 31.83 | 32.93 | 28.64 | 34.97 | 37.11 | 30.61 | 40.66 | 40.94 |
| | Global-Local (Yu et al., 2023) | ViT-B | SAM, CLIP | 33.12 | 36.52 | 29.58 | 35.29 | 39.58 | 31.89 | 40.08 | 40.74 |
| | CaR (Sun et al., 2024b) | ViT-B and ViT-L | CLIP | 33.57 | 35.36 | 30.51 | 34.22 | 36.03 | 31.02 | 36.67 | 36.57 |
| | Ref-Diff (Ni et al., 2023) | ViT-B | SAM, SD, CLIP | 37.21 | 38.40 | 37.19 | 37.29 | 40.51 | 33.01 | 44.02 | 44.51 |
| | TAS (Suo et al., 2023) | ViT-B | SAM, BLIP2, CLIP | 39.84 | 41.08 | 36.24 | **43.63** | **49.13** | 36.54 | 46.62 | 46.80 |
| | HybridGL (Liu & Li, 2025) | ViT-B | SAM, CLIP | 49.48 | 53.37 | 45.19 | 43.40 | **49.13** | 37.17 | **51.25** | **51.59** |
| | REFAM (ours) | DiT | SAM, FLUX | **57.24** | **59.78** | **53.32** | 43.59 | 47.28 | **38.77** | 47.11 | 48.35 |

Table 2: **Comparison with state-of-the-art zero-shot methods on RefCOCO, RefCOCO+, and RefCOCOg.** The top two results in each setting (without additional training) are marked in **bold** and underlined, respectively. * denotes use of extra training data beyond the task-specific set.

| Method | Ref-DAVIS17 $\mathcal{J}\&\mathcal{F}$ | Ref-DAVIS17 $\mathcal{J}$ | Ref-DAVIS17 $\mathcal{F}$ | Ref-YouTube-VOS $\mathcal{J}\&\mathcal{F}$ | Ref-YouTube-VOS $\mathcal{J}$ | Ref-YouTube-VOS $\mathcal{F}$ | MeViS $\mathcal{J}\&\mathcal{F}$ | MeViS $\mathcal{J}$ | MeViS $\mathcal{F}$ |
|---|---|---|---|---|---|---|---|---|---|
| | | | Training-Free with Grounded-SAM | | | | | | |
| Grounded-SAM (Ren et al., 2024)† | 65.2 | 62.3 | 68.0 | 62.3 | 61.0 | 63.6 | - | - | - |
| Grounded-SAM2 (Ren et al., 2024)† | 66.2 | 62.6 | 69.7 | 64.8 | 62.5 | 67.0 | 38.9 | 35.7 | 42.1 |
| AL-Ref-SAM2 (Huang et al., 2025) | 74.2 | 70.4 | 78.0 | 67.9 | 65.9 | 69.9 | 42.8 | 39.5 | 46.2 |
| | | | **Training-Free** | | | | | | |
| G-L + SAM2  (Yu et al., 2023)† | 40.6 | 37.6 | 43.6 | 27.0 | 24.3 | 29.7 | 23.7 | 20.4 | 30.0 |
| G-L (SAM) + SAM2  (Yu et al., 2023)† | 46.9 | 44.0 | 49.7 | 33.6 | 29.9 | 37.3 | 26.6 | 22.7 | 30.5 |
| REFAM + SAM2 (ours) | **57.6** | **54.5** | **60.6** | **42.7** | **37.6** | **47.8** | **30.6** | **24.7** | **36.6** |

Table 3: **Comparison with state-of-the-art zero-shot methods Ref-DAVIS17, Ref-YouTube-VOS and MeViS.** † Results are from Al-Ref (Huang et al., 2025).

**Inference Efficiency.** While our method utilizes large diffusion backbones, it avoids the complex auxiliary modules found in prior works. For instance, HybridGL (Liu & Li, 2025) relies on multiple inference passes and proposal networks, resulting in a reported total inference time of ∼1.1 seconds per image. In contrast, RefAM requires approximately 460ms per image (using FLUX-dev on an A100 GPU), making it significantly faster than the strongest training-free baselines while achieving higher accuracy. Memory usage (∼22GB) remains within standard research hardware limits for large-scale foundation models.

## 4.2 ABLATIONS

In Tabs. 4 and 5, we decouple $\mathcal{J}\&\mathcal{F}$ mask evaluation from our predicted points by introducing the point accuracy (PA) metric, which considers a point correct if it falls within the ground-truth mask.

**Influence of Attention Magnets.** Including stop words in attention map aggregation results in overly diffuse localization. As shown in Tabs. 4 and 6, introducing and then filtering our attention magnets (AM) out from the referring expressions improves predicted point accuracy from 59.9 to 68.9 and raises the $\mathcal{J}\&\mathcal{F}$ metric by 3.2 points on RVOS. Moreover, we observe consistent gains across settings when attention magnets are appended. Fig. 5 further illustrates how redistributing background activations followed by filtering, produces sharper and more focused attention maps.

| AM | NP | SB | Ref-DAVIS17 | | | |
|----|----|----|------|------|------|------|
| | | | $\mathcal{J}\&\mathcal{F}$ | $\mathcal{J}$ | $\mathcal{F}$ | PA |
| ✔ | ✔ | ✔ | 57.6 | 54.5 | 60.6 | 68.9 |
| - | ✔ | ✔ | 54.4 | 50.9 | 57.6 | 59.8 |
| ✔ | ✔ | - | 55.1 | 52.2 | 58.0 | 67.2 |
| - | ✔ | - | 53.1 | 49.5 | 56.7 | 60.2 |
| ✔ | - | - | 54.2 | 51.5 | 56.9 | 59.0 |
| - | - | - | 50.0 | 46.8 | 53.2 | 52.5 |

Table 4: **Influence of the components.** AM denotes appending attention magnets with the following filtering, NP is filtering of everything but noun phrase, SB is spatial bias. PA is predicted point accuracy.

| AM | Ref-DAVIS17 | | | |
|----|------|------|------|------|
| | $\mathcal{J}\&\mathcal{F}$ | $\mathcal{J}$ | $\mathcal{F}$ | PA |
| stop words + color | 57.6 | 54.5 | 60.6 | 68.9 |
| stop words | 56.8 | 53.7 | 59.9 | 67.2 |
| random stop words (5x) | 57.5 | 54.3 | 60.5 | 68.5 |
| random vectors (5x) | 56.2 | 53.1 | 59.4 | 65.5 |
| none | 54.4 | 50.9 | 57.6 | 59.8 |
| scene description | 48.9 | 45.2 | 52.2 | 60.6 |

Table 5: **Influence of different AM.** AM denotes appending attention magnets to the referral expression. PA is predicted point accuracy.

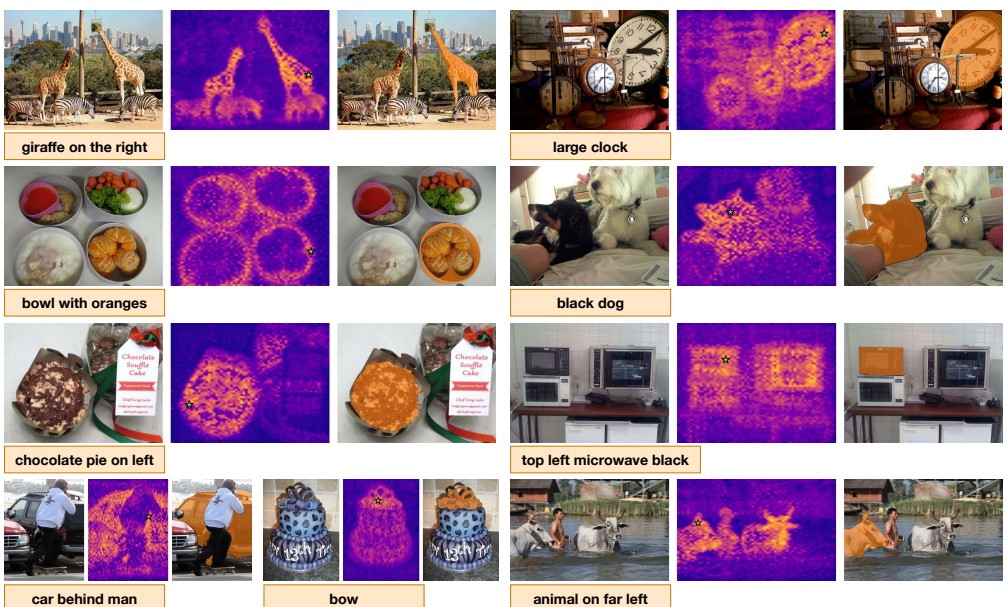

Figure 6: **Qualitative examples.** Referring image object segmentation results. Each triplet shows the input image with the referring expression, the cross-attention heatmap with the detected argmax point (star), and the final segmentation mask produced by SAM.

| AM | NP | SB | RefCOCO | | | RefCOCO+ | | | RefCOCOg | |
|----|----|----|------|-------|-------|------|-------|-------|------|------|
| | | | val | testA | testB | val | testA | testB | val | test |
| ✔ | ✔ | ✔ | 46.91 | 52.30 | 43.88 | 38.57 | 42.66 | 34.90 | 45.53 | 44.45 |
| - | ✔ | ✔ | 33.89 | 44.66 | 34.14 | 35.12 | 37.69 | 33.75 | 42.93 | 42.44 |
| ✔ | ✔ | - | 37.60 | 41.81 | 34.22 | 38.53 | 42.60 | 35.66 | 42.59 | 42.75 |
| - | ✔ | - | 32.99 | 35.22 | 31.98 | 34.47 | 37.05 | 33.62 | 41.83 | 41.00 |
| ✔ | - | - | 35.54 | 39.80 | 32.86 | 37.12 | 40.89 | 34.32 | 38.66 | 40.90 |
| - | - | - | 29.14 | 31.49 | 29.21 | 31.61 | 34.29 | 30.65 | 34.09 | 35.81 |

Table 6: Ablation on spatial bias and noun phrase encoding. Both components contribute to performance, with spatial bias providing the largest gains, while combining both yields the best results across RefCOCO, RefCOCO+, and RefCOCOg.

**Variants of Attention Magnets.** In Tab. 5, we evaluate the role of including color tokens as attention magnets. As discussed above, they help redistribute GAS away from meaningful tokens in the referring expressions, yielding an improvement of roughly 1% across metrics. We then

examine whether the specific choice of stop words matters. Sampling five different stop-word sets produces consistent results. However, replacing stop words with random vectors (re-normalized to match token distributions) leads to slightly worse performance. This suggests that background redistribution is crucial for capturing semantics in generative models, and that real stop words, which are frequently encountered during training, are particularly effective at absorbing meaningless background activations.

**Noun Prase and Spatial Bias.** We conduct an ablation study to disentangle the contributions of spatial bias and noun phrase encoding, as shown in Tab. 6 and Tab. 4. We use the same preprocessing strategy as HybridGL (Liu & Li, 2025) to extract noun phrases and spatial relations from the referring expression, ensuring a fair comparison. When combined, the two components with our attention magnets yield the best performance across all benchmarks, confirming their complementary roles in grounding referring expressions. See Sec. C for more details.

**Qualitative Examples.** Figs 6 and 5 present qualitative examples of RIOS and RVOS. Each example shows the input image, the corresponding cross-attention map with the predicted argmax location indicated by a star, and the final segmentation mask produced by SAM when seeded with this location. Fig. 5 additionally shows aggregated attention maps with and without our attention magnets. We observe that REFAM accurately grounds diverse referring expressions including various attributes. While the attention maps often highlight multiple candidate regions when objects are visually similar, the predicted argmax location reliably falls on the correct instance, enabling accurate segmentation.

### 4.3 GENERALIZATION & BACKBONE ANALYSIS

To verify that our performance gains stem from the proposed methodology rather than solely from the specific FLUX backbone, we extend our evaluation to Stable Diffusion 3.5 (SD3.5). SD3.5 employs a hybrid text encoding scheme utilizing both CLIP and T5 encoders, allowing us to decouple their contributions and analyze the source of semantic grounding.

**T5 vs. CLIP Structure.** As shown in Table 7, utilizing the T5 encoder alone yields significantly better performance than CLIP. We observe that T5 is structure-aware: removing stop words (w/o RefAM) causes a sharp performance drop (e.g., -10.9% mIoU on RefCOCO Test A). Conversely, CLIP acts effectively as a "bag-of-words" model (Yuksekgonul et al., 2023); removing stop words often improves its performance, indicating it fails to utilize syntactic structure for fine-grained grounding. This validates our design choice: RefAM exploits the fine-grained structural alignment present in modern T5-based DiTs, which is largely absent in CLIP-based dual-encoders. We provide the complete evaluation across all datasets in the Appendix.

| Metric | T5 Encoder | | CLIP Encoder | | Combined | |
|---|---|---|---|---|---|---|
| | w/ RefAM | w/o RefAM | w/ RefAM | w/o RefAM | w/ RefAM | w/o RefAM |
| oIoU | **38.4** | 27.2 | 37.1 | 35.5 | 38.5 | 36.0 |
| mIoU | **45.8** | 33.3 | 44.2 | 41.1 | 46.7 | 41.6 |

Table 7: Backbone Analysis on SD3.5 (RefCOCO Test A). T5 provides structural understanding (sensitive to Stop Words), while CLIP behaves like a Bag-of-Words.

## 5 CONCLUSION

We introduce REFAM, a training-free framework for zero-shot referring segmentation that exploits cross-attention features from flow-matching DiTs. By identifying stop words as attention magnets and uncovering global attention sinks (GAS), we proposed a simple redistribution mechanism that sharpens localization without retraining or architectural changes. REFAM sets a new state of the art among training-free methods: on RefCOCO, RefCOCO+, and RefCOCOg it outperforms previous zero-shot approaches, including gains of up to +2.5 mIoU over HybridGL, and on Ref-DAVIS17, Ref-YouTube-VOS, and MeViS it achieves the best reported results for video. These findings highlight diffusion attention as a powerful, general foundation for grounding referring expressions in both images and videos.

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

APPENDIX CONTENTS

## Use of Large Language Models (LLMs)

We used an LLM as an auxiliary tool in two limited capacities. For dataset preparation, the model generated captions for DAVIS, Ref-YouTube-VOS, and MeViS test videos, as well as semantically related prompt variations. All generated outputs were carefully screened and curated by the authors before use. For writing, the authors drafted all sections of the paper, and the LLM was employed only for copy-editing and stylistic refinement. The model was never used to introduce technical content, conduct experiments, or contribute to the research methodology.

## A  Additional Experiments

### A.1  Full Backbone Analysis on SD3.5

In the main text, we presented the ablation study on the RefCOCO TestB split to demonstrate the structural differences between T5 and CLIP encoders. In Table 8, we provide the complete evaluation across all splits of RefCOCO, RefCOCO+, and RefCOCOg.

The results consistently confirm our findings: the T5 encoder is structure-aware and suffers significant performance drops when stop words are removed (w/o RefAM). In contrast, the CLIP encoder acts largely as a "bag-of-words" model, often showing insensitivity or even slight improvements when stop words are removed, but failing to achieve the peak performance of the T5 encoder on complex splits.

Table 8: Full ablation of Text Encoders in SD3.5 across RefCOCO, RefCOCO+, and RefCOCOg (Metric: oIoU). T5 consistently outperforms CLIP and is sensitive to RefAM (Stop Words), confirming it drives structural grounding.

| Dataset | Split | T5 (Structure) | | CLIP | | T5+CLIP | |
|---|---|---|---|---|---|---|---|
| | | w/ RefAM | w/o RefAM | w/ RefAM | w/o RefAM | w/ RefAM | w/o RefAM |
| RefCOCO | Val | 35.2 | 26.5 | 32.5 | 32.2 | 34.5 | 31.9 |
| | TestA | 38.4 | 27.2 | 37.1 | 35.5 | 38.5 | 36.0 |
| | TestB | 34.3 | 25.9 | 27.0 | 30.5 | 28.4 | 30.2 |
| RefCOCO+ | Val | 27.8 | 22.3 | 28.6 | 25.2 | 28.7 | 24.9 |
| | TestA | 30.2 | 23.2 | 31.4 | 27.5 | 31.8 | 27.8 |
| | TestB | 25.0 | 21.0 | 24.2 | 23.1 | 24.2 | 21.7 |
| RefCOCOg | Val | 27.6 | 25.7 | 26.6 | 25.8 | 27.0 | 24.5 |
| | Test | 28.0 | 27.2 | 27.8 | 26.3 | 28.0 | 25.9 |

### A.2  Feature Collection with Inversion

In the main paper, we extract diffusion features without applying inversion. For completeness, we also report results obtained when performing inversion prior to feature collection.

**Setup.** All experiments were conducted on a single NVIDIA A100 GPU. The peak memory usage was approximately 37GB for both Flux and Mochi models. For REFAM + inversion, the inversion step requires 6–9 seconds per input, while subsequent denoising (28 steps) requires 16–20 seconds. Video segmentation with 50 frames using SAM2 adds 3–4 seconds. We use a batch size of 1 for the visual input, but our framework supports multiple referring expressions per image or video. In practice, we process 5–8 expressions in parallel without a hard limit. Overall, feature extraction with REFAM + inversion is roughly 60% faster than the commonly used SD+DINO pipeline.

**Feature collection via inversion.** Generative diffusion models are trained to synthesize images from pure noise, and directly perturbing a real image with Gaussian noise does not guarantee that the resulting latent follows a trajectory observed during generation. This mismatch often causes reconstructions that diverge from the input in content or style (Hertz et al., 2022; Rout et al., 2025). To address this, we adopt rectified flow inversion (Rout et al., 2025), which introduces a controlled backward process ensuring trajectory consistency. The latent dynamics are defined as

$$\frac{dY_t}{dt} = -v_{1-t}(Y_t) + \gamma\big(u_t(Y_t \mid y_1) + v_{1-t}(Y_t)\big),$$

where $u_t(\cdot)$ is the forward vector field, $v_{1-t}(\cdot)$ its reverse, and $\gamma \in [0, 1]$ a correction factor. This inversion yields structured noise latents $X_T$ that, when denoised, reconstruct the original input $X_0$ while preserving semantic fidelity.

**Procedure.** Our feature collection consists of two stages: (i) invert the clean input $X_0$ into a structured noise latent $X_T$ via rectified flow inversion; (ii) apply standard rectified flow denoising to $X_T$, collecting cross-attention maps from intermediate DiT blocks. To further improve semantic quality, we generate a caption for each input and condition both inversion and denoising on this caption. This ensures that the extracted features encode meaningful semantics aligned with downstream referring segmentation tasks.

**Results and trade-off.** We compare feature extraction with and without inversion on standard referring expression benchmarks in Table 9. While both variants establish new state-of-the-art zero-shot results, inversion consistently improves performance across RefCOCO, RefCOCO+, and RefCOCOg, particularly in terms of mIoU. However, these gains come at the cost of additional latency: inversion adds 6–9 seconds per input before denoising. Thus, practitioners may choose between the non-inversion variant for efficiency, or the inversion-based variant for maximal accuracy, depending on application requirements.

| Metric | Method | Vision Backbone | Pre-trained Model | RefCOCO | | | RefCOCO+ | | | RefCOCOg | |
| --- | --- | --- | --- | --- | --- | --- | --- | --- | --- | --- | --- |
| | | | | val | testA | testB | val | testA | testB | val | test |
| oIoU | REFAM (ours) | DiT | SAM, FLUX | 46.91 | **52.30** | 43.88 | 38.57 | 42.66 | 34.90 | 45.53 | 44.45 |
| | REFAM + Inversion (ours) | DiT | SAM, FLUX | **47.99** | 52.22 | **43.99** | **40.29** | **46.05** | **36.71** | **47.08** | **47.38** |
| mIoU | REFAM (ours) | DiT | SAM, FLUX | 57.24 | 59.78 | 53.32 | 43.59 | 47.28 | 38.77 | 47.11 | 48.35 |
| | REFAM + Inversion (ours) | DiT | SAM, FLUX | **59.12** | **60.21** | **55.02** | **46.86** | **51.13** | **42.30** | **49.42** | **50.55** |

Table 9: **Zero-shot referring segmentation results on RefCOCO, RefCOCO+, and RefCOCOg.** We compare our method with and without inversion. Both variants achieve state-of-the-art performance, and inversion further improves consistency and semantic alignment, particularly in mIoU. Best and second-best results are shown in **bold** and underlined, respectively.

## A.3 REFERRAL VIDEO OBJECT SEGMENTATION

**Representation Space.** In Tab. 10, we compare cross-attention maps with output representations of attention. The output space is used in Concept Attention (CA) (Helbling et al., 2025), which has been shown to perform better for single-object segmentation tasks. However, a key limitation of CA is its reliance on a predefined set of simple, one-word concepts to represent the entire scene. For example, to segment an image of a dragon sitting on a stone, concepts like "dragon", "rock", "sun", and "clouds" must all be explicitly defined. In contrast, our approach detects references without requiring detailed scene decomposition and instead relies solely on multi-word, complex concepts defined by the referring expression. We observe that cross attention representation space shows better results than the proposed attention output in CA (Helbling et al., 2025).

| Space | Ref-DAVIS17 | | | |
| --- | --- | --- | --- | --- |
| | $\mathcal{J}\&\mathcal{F}$ | $\mathcal{J}$ | $\mathcal{F}$ | PA |
| Attention Output | 55.6 | 52.3 | 58.8 | 64.8 |
| Cross Attention | 57.6 | 54.5 | 60.6 | 68.9 |

Table 10: **Ablations of representation space.** PA is predicted point accuracy.

**Ablation on Text Conditioning.** We investigate how textual prompting affects performance by varying the use of captions and empty prompts during the reconstruction stages. As shown in Tab. 11, using captions achieves better performance across all metrics. Removing captions results in noticeable performance drops. These results demonstrate that textual prompts are beneficial for the feature extraction from the diffusion models.

**SAM2 Variant.** Finally, we evaluate the effect of using the smaller variant of SAM2. Replacing SAM2-H with SAM2-S leads to a performance decrease across all scores, including a sharp drop in $\mathcal{J}\&\mathcal{F}$ from 57.6 to 51.8. This suggests that higher model capacity is important for capturing fine-grained spatial details in referring video segmentation.

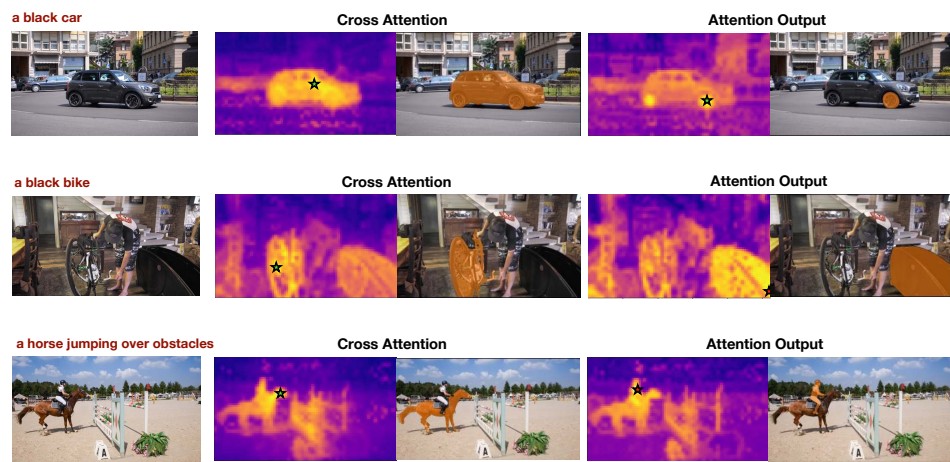

Figure 7: **Visualization of different representaiton spaces.** REFAM features with cross attention representations or output representations of attention. For referral tasks, REFAM use cross attention.

| text condition | Ref-DAVIS17 | | | |
|---|---|---|---|---|
| | $\mathcal{J}\&\mathcal{F}$ | $\mathcal{J}$ | $\mathcal{F}$ | PA |
| empty | 56.6 | 53.5 | 59.8 | 65.5 |
| caption | 57.6 | 54.5 | 60.6 | 68.9 |

Table 11: **Ablations of text conditioning.** PA is predicted point accuracy.

## A.4 COMPARISON TO CONCEPTATTENTION ON IMAGE SEGMENTATION TASK

We compare our method with ConceptAttention (CA) (Helbling et al., 2025) on direct image segmentation using Pascal VOC (Everingham et al., 2015) and ImageNet Segmentation (Guillaumin et al., 2014). While CA supports multi-object segmentation, it requires that all relevant concepts in the scene be explicitly specified in advance. This reliance on a predefined set of simple, often one-word concepts makes it less flexible in open-world or complex scenes, where full concept enumeration is impractical or ambiguous.

In contrast, our method bypasses this requirement by leveraging extra stop words (see Sec. B) that serve as background-attention magnets within cross-attention maps. Additionally, we condition feature extraction on a general caption of the input image, which improves detection performance. This enables segmentation from expressive, natural language descriptions without concept-by-concept supervision. As shown in Tab. 13, our training-free approach performs competitively with CA, and qualitative results in Fig. 7 highlight improved object coverage. Whereas CA often attends to isolated, salient object parts, our method tends to capture the full spatial extent of the described object.

It is also worth noting that CA was originally introduced as an interpretability method for analyzing attention in diffusion transformers, rather than as a practical segmentation technique. In CA, features are extracted from the attention layers of multi-modal DiTs without modifying the denoising trajectory: the model is conditioned on either the source prompt or an empty prompt, and additional concept tokens are introduced only for interpretability. These tokens participate in attention to produce contextualized representations, but do not influence the visual stream or alter the generated image. ConceptAttention saliency maps are then constructed by projecting image patch outputs onto concept embeddings across multiple layers.

By contrast, our approach uses cross-attention features linked to referring expressions and augmented stop words explicitly for segmentation guidance. Thus, while CA provides insight into model internals, our method turns attention mechanisms into a practical tool for zero-shot segmentation via semantic grounding.

| size of SAM | Ref-DAVIS17 | | | |
| | $\mathcal{J}\&\mathcal{F}$ | $\mathcal{J}$ | $\mathcal{F}$ | PA |
| --- | --- | --- | --- | --- |
| small | 51.8 | 48.4 | 55.3 | 68.9 |
| huge | 57.6 | 54.5 | 60.6 | 68.9 |

Table 12: **Influence of size of SAM on RVOS.** PA is predicted point accuracy.

| Method | Architecture | ImageNet-Segmentation | | | PascalVOC (Single Class) | | |
| | | Acc ↑ | mIoU↑ | mAP↑ | Acc ↑ | mIoU↑ | mAP↑ |
| --- | --- | --- | --- | --- | --- | --- | --- |
| LRP (Binder et al., 2016) | CLIP ViT | 51.09 | 32.89 | 55.68 | 48.77 | 31.44 | 52.89 |
| Partial-LRP (Binder et al., 2016) | CLIP ViT | 76.31 | 57.94 | 84.67 | 71.52 | 51.39 | 84.86 |
| Rollout (Abnar & Zuidema, 2020) | CLIP ViT | 73.54 | 55.42 | 84.76 | 69.81 | 51.26 | 85.34 |
| ViT Attention (Dosovitskiy et al., 2021) | CLIP ViT | 67.84 | 46.37 | 80.24 | 68.51 | 44.81 | 83.63 |
| GradCAM (Selvaraju et al., 2020) | CLIP ViT | 64.44 | 40.82 | 71.60 | 70.44 | 44.90 | 76.80 |
| TextSpan (Gandelsman et al., 2024) | CLIP ViT | 75.21 | 54.50 | 81.61 | 75.00 | 56.24 | 84.79 |
| TransInterp (Chefer et al., 2021) | CLIP ViT | 79.70 | 61.95 | 86.03 | 76.90 | 57.08 | 86.74 |
| DINO Attention (Caron et al., 2021) | DINO ViT | 81.97 | 69.44 | 86.12 | 80.71 | 64.33 | 88.90 |
| DAAM (Tang et al., 2022) | SDXL UNet | 78.47 | 64.56 | 88.79 | 72.76 | 55.95 | 88.34 |
| DAAM (Tang et al., 2022) | SD2 UNet | 64.52 | 47.62 | 78.01 | 64.28 | 45.01 | 83.04 |
| Flux Cross Attention (Helbling et al., 2025) | Flux DiT | 74.92 | 59.90 | 87.23 | 80.37 | 54.77 | 89.08 |
| ConceptAttention (Helbling et al., 2025) | Flux DiT | 83.07 | 71.04 | **90.45** | 87.85 | 76.45 | 90.19 |
| REFAM (ours) | Flux DiT | 85.61 | 71.37 | 87.94 | 89.14 | 78.57 | 90.09 |
| REFAM (ours) + Inversion | Flux DiT | **87.28** | **73.33** | 90.21 | **89.82** | **80.07** | **90.70** |

Table 13: Our method, REFAM with inversion, consistently outperforms a range of interpretability techniques based on Diffusion, DINO, CLIP ViT, and Flux DiT on both ImageNet-Segmentation and PascalVOC (Single Class). The performance numbers for the other methods are taken directly from ConceptAttention, and we follow the same evaluation procedure to ensure fair comparison.

## B    STOP WORD FILTERING & ADDITIONAL STOP WORDS

In this section, we discuss the rationale behind filtering stop words from the attention maps and describe the method we employ to accomplish this.

**Stop Word Filtering.**    Given a referral expression $e$ tokenized into $K$ tokens $\{t_k\}_{k=1}^{K}$, and an input image or video, we compute cross-attention maps between each text token $t_k$ and all the visual tokens in the image or video frames. Consequently, for each token $t_k$, there exists a corresponding cross-attention map $H_k$.

To normalize these attention maps, we apply a softmax function across all tokens:

$$\hat{H}_k = \text{softmax}_k(H_k).$$

This normalization implies that for each visual patch we define a probability distribution that associate it with the token having the highest softmax score relative to that patch. Given that the referral expression corresponds specifically to a particular region or element within the visual input, it follows that visual areas not directly associated with the referral expression must be attributed to other tokens. We observe that words with minimal semantic significance, such as stop words, often represent the broader context or background elements of the scene relative to the specific referral expression.

Observing this behavior, we propose to filter out attention maps corresponding to stop words before averaging attention maps, resulting in more focused and precise attention representations of the referral expression.

See Fig. 11, Fig. 12, Fig. 13, and Fig. 14 for qualitative examples illustrating attention maps per token associated with stop words.

**Extra Stop Words.**    We observe that the given referral expression $e$ usually contains a limited number of stop words, insufficient to effectively capture all background details of the visual input. To allow finer granularity in attention-to-token associations, we introduce additional stop words that act as magnets for background attention during the softmax computation. Similarly to the existing stop

words in the referral expression, we filter out these attention maps associated with additional stop words after computing the softmax and before averaging the attention maps.

See Fig. 11, Fig. 12, Fig. 13, and Fig. 14 for comparisons of attention maps calculated with and without extra stop words.

**List of Stop Words.** Below we list stop words that we filter during attention computation semicolon separated. The stop words are taken from NLTK library and extended by symbols "_", ",", "." to account for the special symbols from the tokenization.

```
i; me; my; myself; we; our; ours; ourselves; you;
your; yours; yourself; yourselves; he; him; his;
himself; she; her; hers; herself; it; its; itself;
they; them; their; theirs; themselves; what; which;
who; whom; this; that; these; those; am; is; are;
was; were; be; been; being; have; has; had; having;
do; does; did; doing; a; an; the; and; but; if; or;
because; as; until; while; of; at; by; for; with;
about; against; between; into; through; during;
before; after; above; below; to; from; up; down; in;
out; on; off; over; under; again; further; then; once;
here; there; when; where; why; how; all; any; both;
each; few; more; most; other; some; such; no; nor;
not; only; own; same; so; than; too; very; s; t; can;
will; just; don; should; now; ,; .; _;
```

## C   DETAILS OF NOUN PHRASE AND SPATIAL BIAS EXTRACTION

We adopt the same preprocessing strategy as HybridGL (Liu & Li, 2025) to extract *noun phrases* and *spatial cues* from referring expressions. Following their approach, we parse each input sentence into object-centric noun phrases (e.g., "the man", "a red car") and spatial relations (e.g., "left of", "behind", "top"). The noun phrases are then encoded with the text encoder and compared against diffusion-derived visual features, guiding attention toward semantically relevant regions.

Spatial cues are incorporated as lightweight spatial priors. Relative relations ("left of the dog") are modeled by comparing bounding box centroids of candidate regions, while absolute terms ("top left", "bottom right") are mapped to normalized positional masks over the image grid. This procedure mirrors the spatial relationship guidance in HybridGL but operates directly on cross-attention features extracted from diffusion transformers.

As shown in our ablation study (Table 6), both components contribute complementary gains. Spatial bias alone improves localization accuracy, while noun phrase extraction enhances semantic alignment. When combined with our attention magnet mechanism, the two yield the strongest results across benchmarks. We further confirm this effect in video segmentation benchmarks (Table 4), where the same preprocessing consistently benefits temporal grounding.

## D   LIMITATIONS

While our approach demonstrates strong performance across referring object segmentation tasks, there are a few aspects that warrant further consideration. The method benefits from high-quality captions, which better guide semantic alignment; when unavailable, we rely on LLM-generated descriptions. Although this introduces a soft dependence on LLMs, performance does not degrade substantially with empty prompts. Moreover, in video referring object segmentation (VROS), we currently ignore temporal aspects of the expression and always localize in the first frame. Future improvements will require detecting the frame in which the referred object actually appears.

Additionally, we use SAM2 (Ravi et al., 2025) to generate a segmentation mask of an object. Generally, SAM2 takes as an input prompt point, multiple points, or a bounding box. In the context of referring image and video segmentation, we use a single point per referring expression. This

approach can lead to undersegmentation, as illustrated in Fig. 8. For instance, even animals may be only partially segmented, only one ear of a camel was segmented in one of the examples. This limitation could be addressed by employing a more sophisticated strategy for sampling points from the output attention maps.

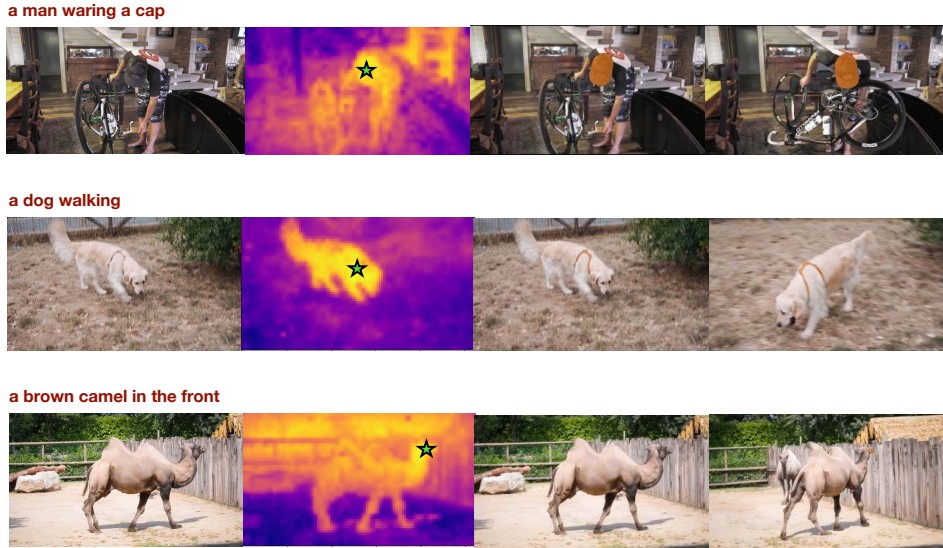

Figure 8: **Visualization of SAM2 failure under-segmentations.**

## E  SOCIETAL IMPACT

Our work presents a training-free framework for referral image and video object segmentation using cross-attention features from large diffusion models. By avoiding task-specific fine-tuning and leveraging existing pre-trained models, our method reduces the need for supervised datasets and extensive retraining. However, it still depends on powerful foundation models, such as FLUX, Mochie and SAM2, trained on large-scale image-text and video-text datasets, the exact composition of which is not always publicly disclosed. As prior studies have shown, large-scale training datasets can contain cultural, racial, or gender biases that may propagate into downstream tasks (Buolamwini & Gebru, 2018; Shankar et al., 2017; Torralba & Efros, 2011). Even in segmentation or correspondence, these biases may lead to varying performance across different demographic groups or underrepresented visual domains (De Vries et al., 2019). Additionally, our reliance on natural language prompts or LLM-generated captions introduces a soft dependence on language models that may encode their own textual biases (Bender et al., 2021; Caliskan et al., 2017). We encourage future work toward training diffusion models on more transparent and carefully curated datasets. However, the considerable computational cost of such efforts continues to pose challenges, especially in academic settings. Our method is intended for research applications such as content-based retrieval, visual understanding, and open-set image analysis, and is not designed for high-risk or sensitive decision-making domains such as surveillance or biometric identification.

## F  QUALITATIVE EXAMPLES

We present qualitative results to illustrate the effectiveness of our method across various referring image and video object segmentation scenarios. These examples highlight how our method, REFAM, captures semantically meaningful regions aligned between object and with the referring expression, and how segmentation quality benefits from attention-based guidance. We also visualize the effect of our stop word filtering strategy, showing improved focus on target objects and reduced attention to irrelevant regions. The following figures show qualitative examples and comparisons across different settings, as shown in Figs. 9 to 14.

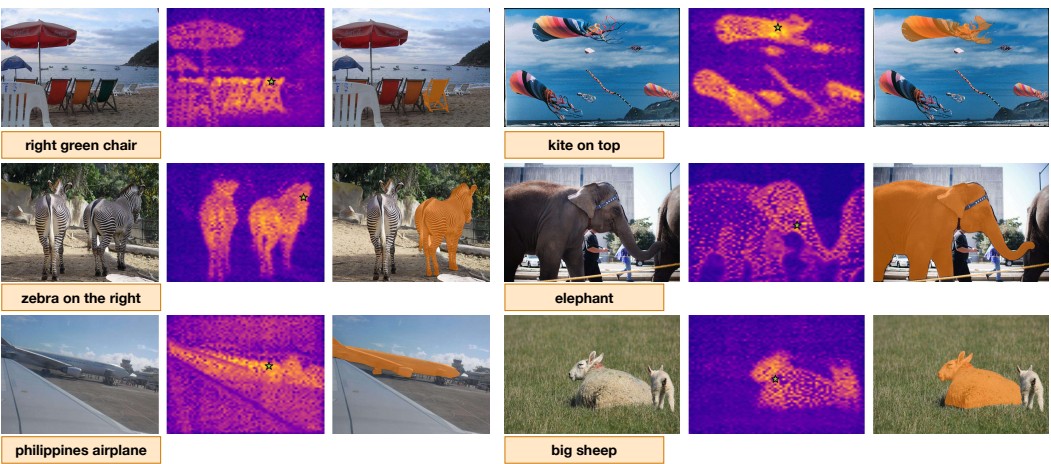

Figure 9: **Additional qualitative examples for referring image object segmentation.** Each panel shows the input image with its corresponding referring expression and the predicted segmentation mask. These examples complement the results in the main paper and illustrate the diversity of object categories and spatial references handled by our method.

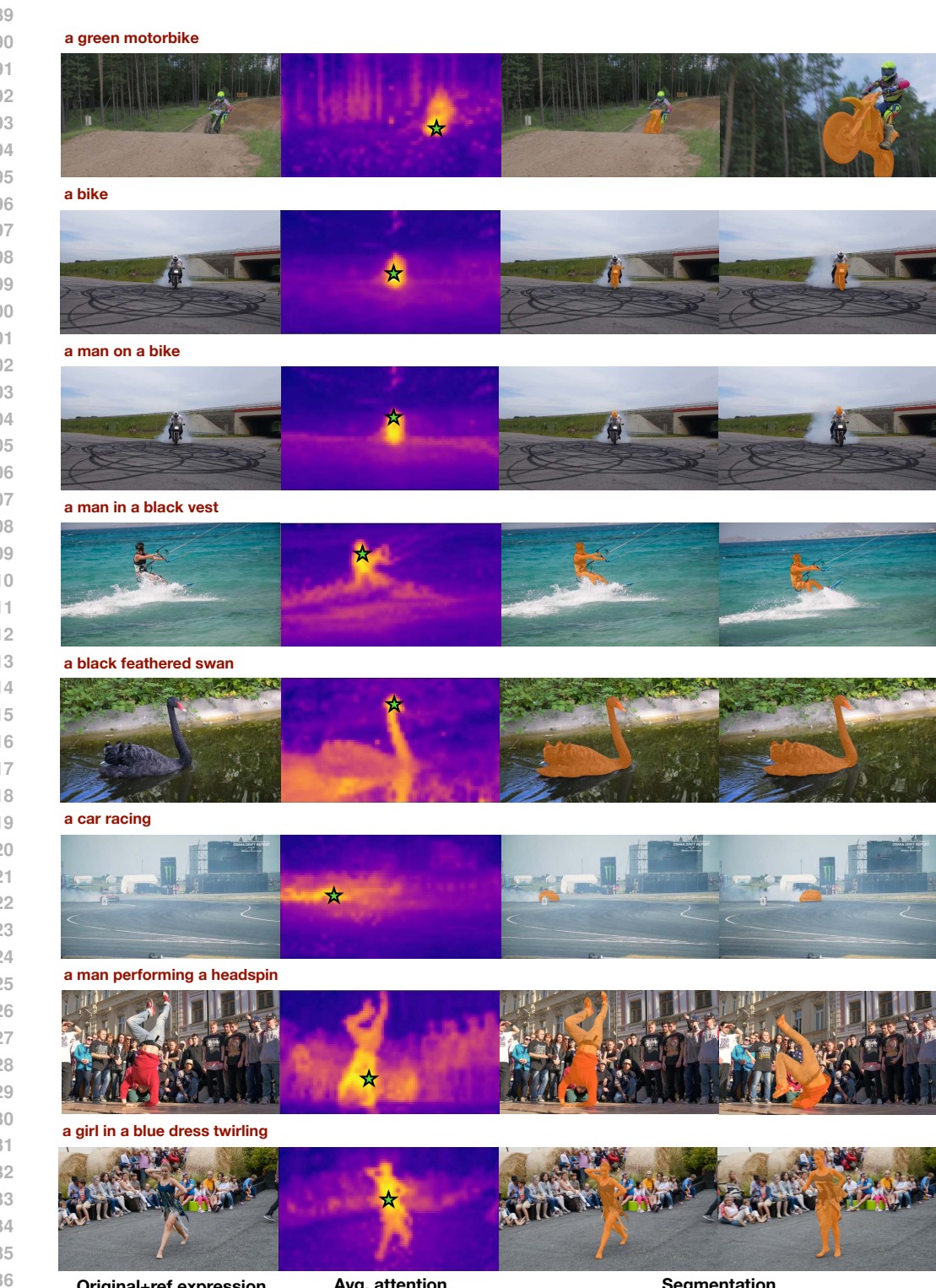

**a green motorbike**

**a bike**

**a man on a bike**

**a man in a black vest**

**a black feathered swan**

**a car racing**

**a man performing a headspin**

**a girl in a blue dress twirling**

Original+ref.expression          Avg. attention          Segmentation

Figure 10: **Qualitative examples.** VROS task, evaluated on Ref-DAVIS17. From left to right: first frame of the video with the corresponding ref.expression on the top, avg. attention map from our REFAM + inversion features, segmentation outputs with SAM2.

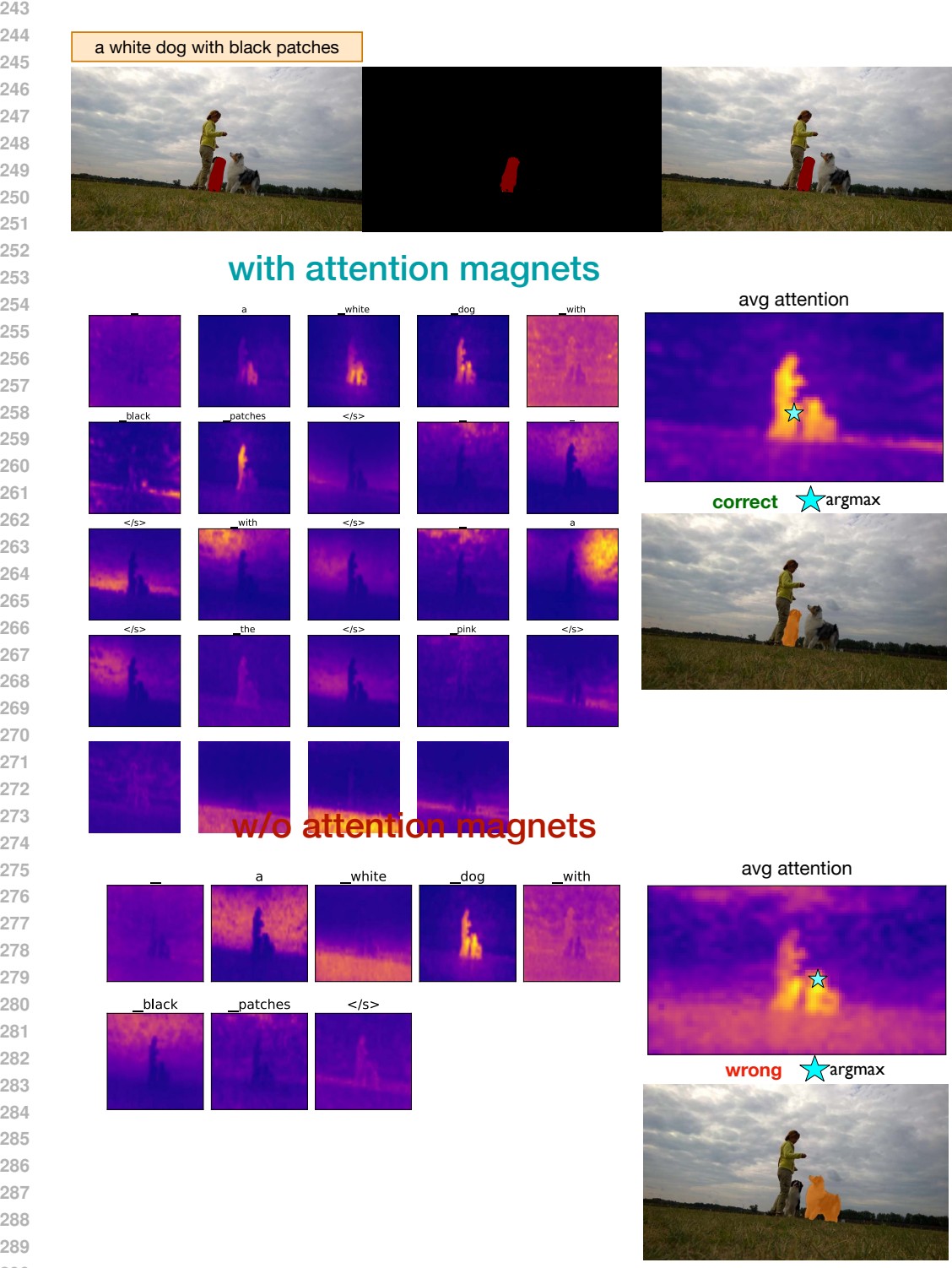

Figure 11: **Qualitative examples.** Qualitative comparison of attention maps obtained with and without additional stop words. The top row shows the first frame of the video along with the corresponding referring expression. The first row includes the average attention map, where the star indicates the argmax point with indication if it was correctly detected.

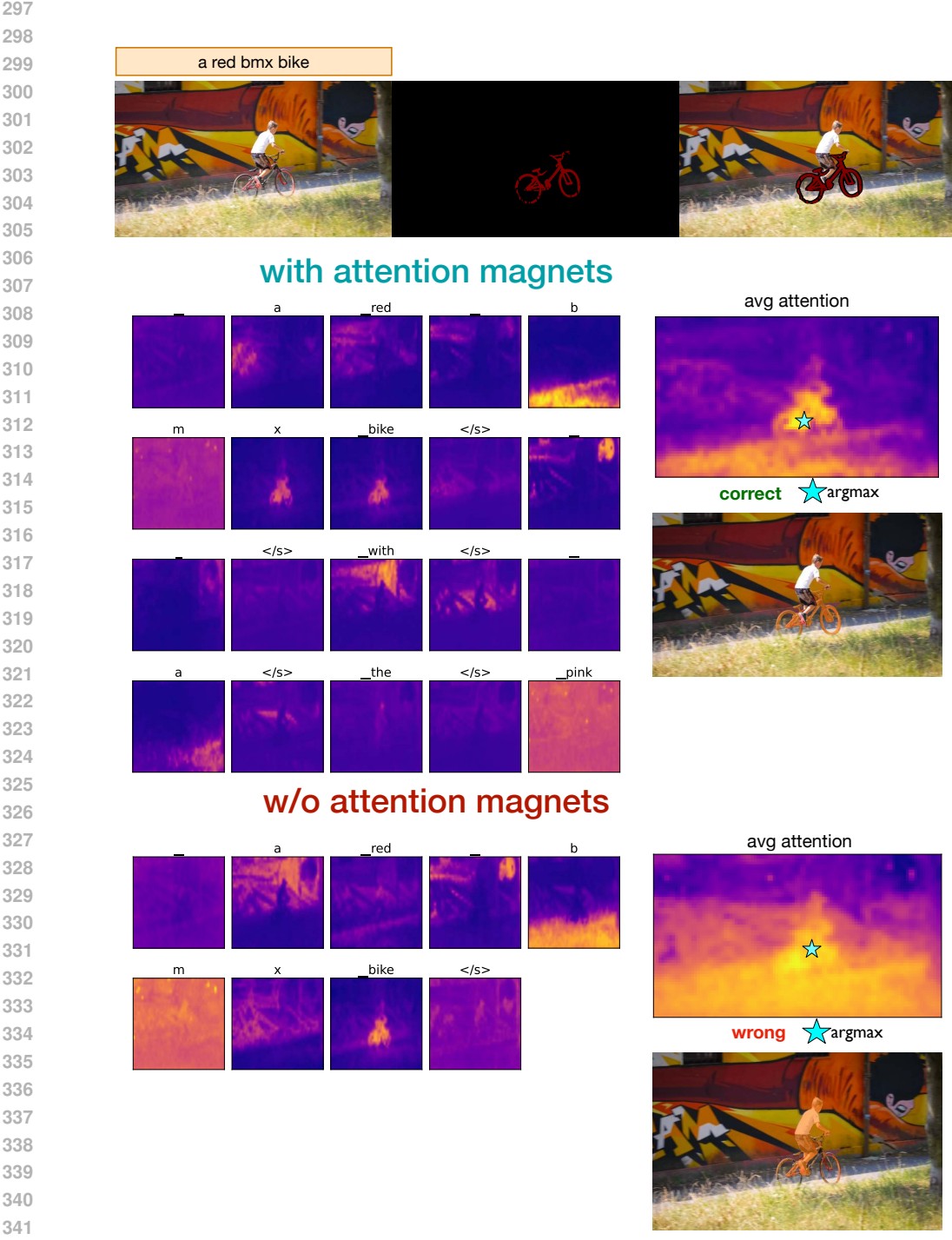

Figure 12: **Qualitative examples.** Qualitative comparison of attention maps obtained with and without additional stop words. The top row shows the first frame of the video along with the corresponding referring expression. The first row includes the average attention map, where the star indicates the argmax point with indication if it was correctly detected.

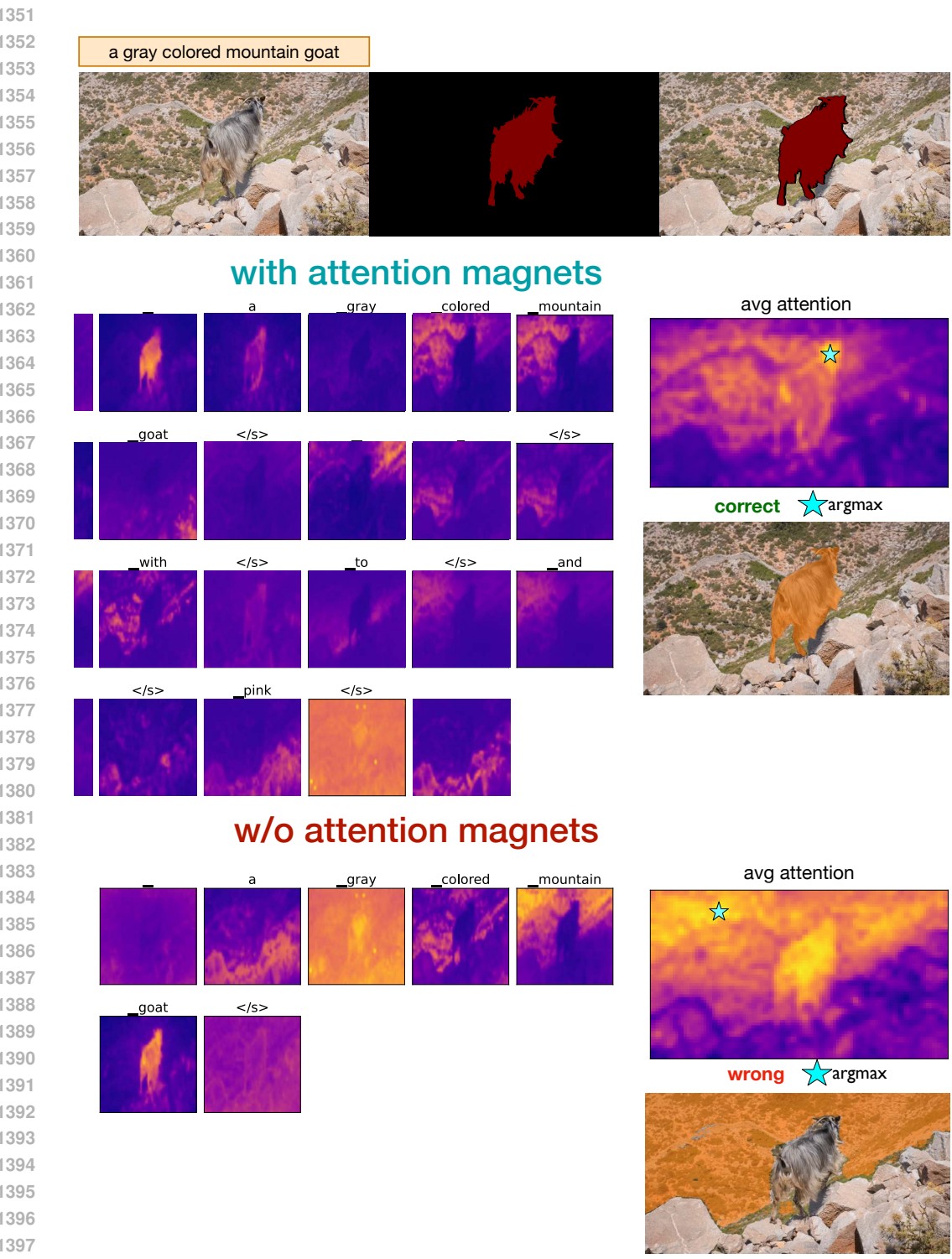

Figure 13: **Qualitative examples.** Qualitative comparison of attention maps obtained with and without additional stop words. The top row shows the first frame of the video along with the corresponding referring expression. The first row includes the average attention map, where the star indicates the argmax point with indication if it was correctly detected.

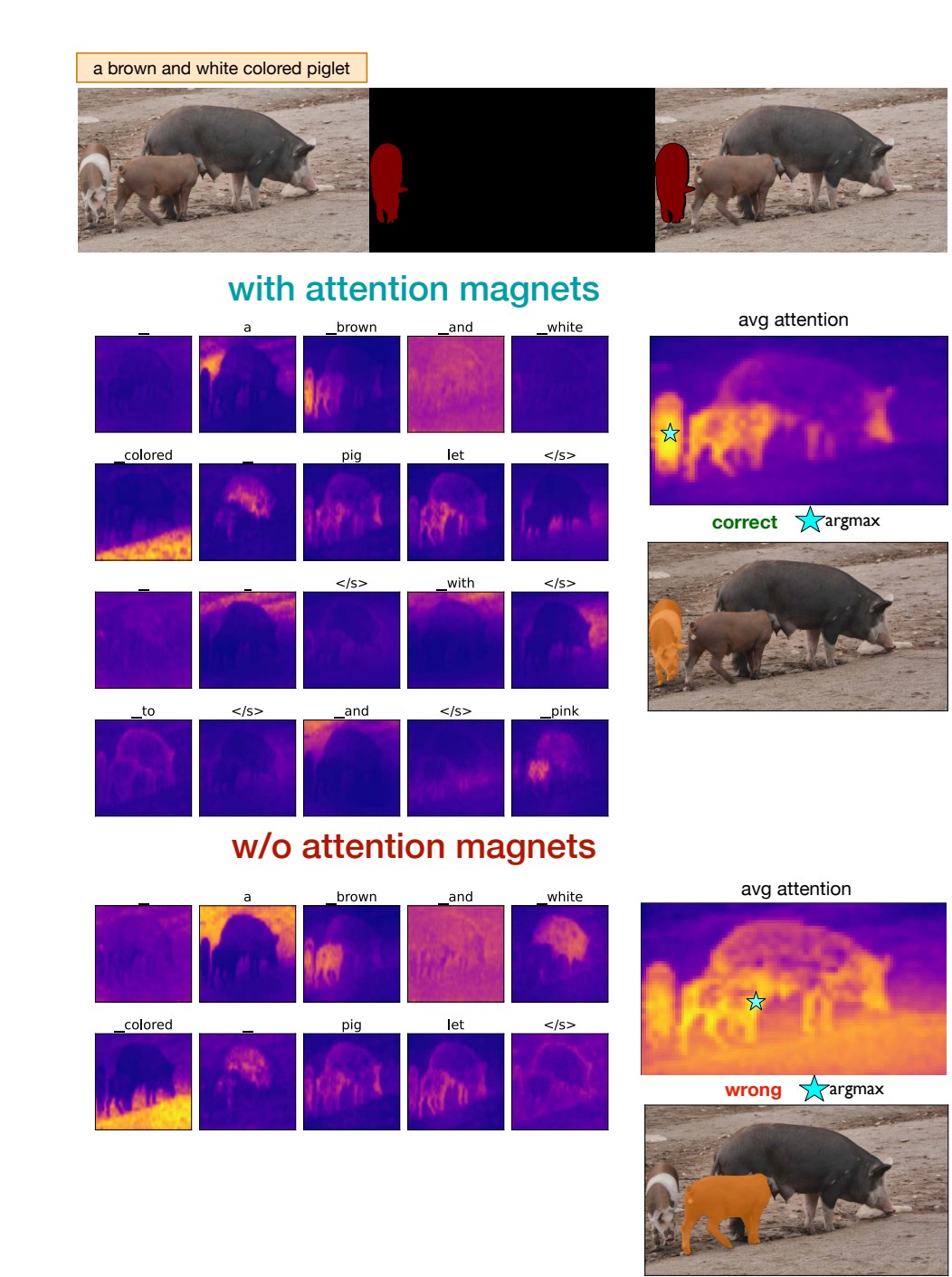

Figure 14: **Qualitative examples.** Qualitative comparison of attention maps obtained with and without additional stop words. The top row shows the first frame of the video along with the corresponding referring expression. The first row includes the average attention map, where the star indicates the argmax point with indication if it was correctly detected.

