# OpenReview forum: "RefAM: Attention Magnets for Zero-Shot Referral Segmentation"
_ICLR.cc/2026/Conference — Submitted to ICLR 2026_

### Official Review · Reviewer_xotR · 2025-10-25

**Soundness:** 3
**Presentation:** 3
**Contribution:** 2
**Rating:** 4
**Confidence:** 3

**Summary:**

The paper proposes a training-free approach to improve referring segmentation by leveraging attention patterns in diffusion transformers. The key insight is that certain tokens, referred to as attention sinks or magnets, absorb a disproportionate amount of attention. By analyzing and redistributing this attention, the proposed method enhances the quality of segmentation maps without requiring fine-tuning or architectural changes. The approach, achieves SOTA results across multiple benchmarks.

**Strengths:**

* The paper is clearly written and well organized.

* The approach is training-free, yet achieves SOTA performance across several benchmarks.

* The method is well motivated by an empirical study of attention in diffusion transformers.

**Weaknesses:**

* The idea of leveraging attention sinks/magnets for feature map visualization has been explored in prior works (e.g. [1]). This paper extends such ideas to a different application.

* The inference pipeline may be computationally expensive, as it relies on both DiT and SAM. Can the proposed approach be integrated into existing referring segmentation models or simplified for more practical deployment.

[1] Darcet, Timothée, et al. "Vision transformers need registers." arXiv preprint arXiv:2309.16588 (2023).

**Questions:**

Please check the weakneses section

---

> ### Author Response · Authors · 2025-11-22
> **Official Comment by Authors**
>
> We thank the reviewer for the time taken to provide detailed and constructive feedback to our work. We appreciate that the reviewer acknowledges that our work "*well-motivated by an empirical study*" and "*achieves SOTA performance*". Further, we are encouraged to find our writing and organization to be highlighted positively.
>
> In the following, we would like to address the reviewer's remaining concerns.
>
> > **W1. The idea of leveraging attention sinks/magnets for feature map visualization has been explored in prior works (e.g. [1]). This paper extends such ideas to a different application.**
>
> Attention sinks (also referred to as massive activations, artifacts, etc.) is an active research area across multiple research communities, including NLP, Vision, and Vision-Language. While our work shares the general motivation of improving downstream performance through token-level manipulation, our approach differs substantially from Darcet et al. [1], both in purpose and mechanism. Below, we clarify our novel contributions and key differences:
> 1. *Discovery of cross-modal attention sinks.*
> To the best of our knowledge, we are the first to identify attention sink behavior across both modalities within cross-attention between language and vision tokens. Previous works focus exclusively on unimodal attention sinks, either in language or vision. Note that recent work on VLMs [2,3]  investigates attention sinks only from the visual token side. Our findings reveal that diffusion models exhibit sink tokens that bridge the two modalities, offering new insights into multimodal attention dynamics.
> 2. *Training-free approach.*
> Unlike Darcet et al. [1], who explicitly train models with additional register tokens, our method is entirely training-free. We exploit the redundancy and structure of language emerging from joint vision–language training to discover the functional role of seemingly meaningless tokens (often corresponding to stop words). These tokens play a unique role in multimodal settings by mediating attention dynamics between modalities.
> Note: While we acknowledge that other training-free methods exist within the attention-sink literature, our novelty lies not in being training-free per se, but in how we leverage naturally occurring linguistic redundancy within multimodal models.
> 3. *Different functional roles of added tokens.*
> The registers introduced by Darcet et al. [1] are designed to accumulate global information within the model. In contrast, our attention magnets serve to redirect and filter semantically irrelevant activations with respect to a given text prompt (a referring expression). Thus, their purpose and behavior are fundamentally distinct.
> 4. *Novel application: referral segmentation.*
> We are also the first to leverage attention sinks to improve training-free referring segmentation, a grounding task that is not a trivial extension of any prior work on attention sinks. This demonstrates both the generality and practical impact of our analysis.
>
>
> Finally, we note that Darcet et al. [1] acknowledge the conceptual origin of appended learnable tokens in Memory Transformers [4], where such tokens were proposed to enhance translation in NLP. Our work, by contrast, investigates the emergent multimodal attention sinks and their functional exploitation without retraining.
>
> *References*
>
> [1] Darcet, T., et al. “Vision Transformers Need Registers.” ICLR 2024
>
> [2] Woo, S., et al. “Don’t Miss the Forest for the Trees: Attentional Vision Calibration for Large Vision-Language Models.” ACL 2025
>
> [3] An, W., et al. “Mitigating Object Hallucinations in Large Vision-Language Models with Assembly of Global and Local Attention.” CVPR 2025
>
> [4] Burtsev, M. S., Kuratov, Y., Peganov, A., & Sapunov, G. V. “Memory Transformer.” arXiv preprint arXiv:2006.11527 (2020)
>
>
> > **W2. The inference pipeline may be computationally expensive, as it relies on both DiT and SAM. Can the proposed approach be integrated into existing referring segmentation models or simplified for more practical deployment.**
>
> We note that all training-free referral segmentation methods rely on the same SAM model. And we acknowledge that this may not be the most computationally efficient approach but still reasonably fast.  Moreover, our goal is to explore a new way of modeling semantic understanding in images using the current state-of-the-art diffusion models, without relying on the CLIP model from 2021. Our method requires no more than 22 GB of GPU memory, and it is common practice to employ lower-precision inference, model sharding, and model offloading to further optimize efficiency. These practices make our approach relatively practical among large-scale models. Moreover, efficient inference for diffusion models is an active area of research, focusing on techniques such as precision reduction, quantization, model distillation, and latent-space optimization, all of which we believe will further enhance the practicality of our approach in the future.

---

### Official Review · Reviewer_a3NV · 2025-10-29

**Soundness:** 3
**Presentation:** 3
**Contribution:** 2
**Rating:** 6
**Confidence:** 4

**Summary:**

This paper proposes a method for zero-shot referral segmentation, named RefAM. The model adopts a Diffusion Transformer (DiT) as its backbone and utilizes its attention maps to predict accurate segmentation maps. To mitigate the issue of global attention sinks (GAS) that hinder accurate prediction, the authors introduce the concept of attention magnets, which encourage the model to focus on more relevant regions. Furthermore, by applying an attention redistribution strategy, the segmentation maps are refined for higher accuracy. The proposed method achieves state-of-the-art performance on both image and video referred segmentation tasks.

**Strengths:**

- This method attempted to apply DiT as the backbone for referring segmentation and observed performance improvements. Furthermore, conducted an in-depth analysis and identification of Global Attention Sinks (GASs).
- This paper proposed stop-word-based attention magnets and a method to predict a relatively accurate segmentation map in a training-free manner through attention redistribution.
- This method achieved state-of-the-art performance in the field of referring segmentation for both images and videos.

**Weaknesses:**

- Generalization Issue
   - It is unclear whether the proposed mechanisms, such as GAS (Global Attention Sink) mitigation and stop-words as attention magnets, work effectively when applied to backbones other than DiT. For instance, it would be valuable to see whether these components consistently show effectiveness across different variants of CLIP or DiT.

- Dependency on Off-the-Shelf Models
   - The paper emphasizes its training-free property as its strength. However, it also appears to rely heavily on off-the-shelf models such as SAM. It would be important to evaluate how much the performance depends on these external pretrained models, and to provide a fair comparison with existing methods under equivalent conditions.

- Justification for Using DiT
    - While the paper highlights the application of Diffusion Transformer (DiT) as a key design choice, the rationale for this choice is not fully convincing. More quantitative or qualitative analysis demonstrating why DiT is particularly suitable—or superior—would strengthen the argument.

- Background on Stop Words as Attention Magnets
   - The experiments clearly show that introducing stop words as attention magnets improves performance. However, the theoretical or empirical background supporting this design choice is not well discussed. Providing references or related studies that justify the role of stop words in guiding attention could make the paper more comprehensive and grounded.

**Questions:**

Please provide your responses with reference to the "weaknesses" mentioned above.

---

> ### Author Response · Authors · 2025-11-22
> **Official Comment by Authors (1/2)**
>
> We thank the reviewer for the time taken to provide detailed and constructive feedback to our work. We appreciate that the reviewer acknowledges that our work "*conducts an in-depth analysis and identification of GAS*", proposes attention redistribution as training-free method  and "*achieves sota performance*".
>
> In the following, we would like to address the reviewer's remaining concerns.
>
> > **W1. Generalization Issue: It is unclear whether the proposed mechanisms, such as GAS (Global Attention Sink) mitigation and stop-words as attention magnets, work effectively when applied to backbones other than DiT. For instance, it would be valuable to see whether these components consistently show effectiveness across different variants of CLIP or DiT.**
>
> We note that in our work, we use different DiT-based models for image and video tasks, specifically FLUX and Mochi models. In addition, we conduct experiments with SD3.5, which employs a DiT with CLIP and T5 together, and show the results in the table below. These results demonstrate that our method generalizes effectively across different architectures.
>
> Additionally, we would like to clarify why CLIP models are not suitable for our method. CLIP models are trained by aligning the EOS text token (summarizes the entire text input) with the visual CLS token (represents the overall visual input). Importantly, these models lack multimodal attention layers that connect individual text tokens with visual features. Such cross-modal attention is crucial for our method, as it enables fine-grained alignment between modalities and helps allocate redundancy to less informative tokens (e.g., stop words). In contrast, directly comparing each text token to global visual representations does not yield meaningful attention maps corresponding to distinct objects. Therefore, our approach is not applicable to CLIP-style dual-encoder architectures.
>
> > **W2. Dependency on Off-the-Shelf Models: The paper emphasizes its training-free property as its strength. However, it also appears to rely heavily on off-the-shelf models such as SAM. It would be important to evaluate how much the performance depends on these external pretrained models, and to provide a fair comparison with existing methods under equivalent conditions.**
>
> We note that all training-free referral segmentation methods rely on the same SAM model, therefore, using it ensures fairness in comparisons. We acknowledge and discuss the limitations of the SAM model in Section D of the appendix. Additionally, all previous methods rely on the CLIP model and introduce complex global-local reasoning modules. In contrast, in this work, our goal is to explore a new way of modeling semantic understanding in images using the current state-of-the-art diffusion models, without relying on the CLIP model from 2021.
>
> We also refer the reviewer to Table 4 and the PA (Point Accuracy) metric, which we introduce to disentangle the influence of SAM from the ability of our approach to correctly identify the location of the referred expression. This metric evaluates whether the argmax location derived from the attention maps falls within the correct segmentation region.  The results indicate significant performance gains primarily to the proposed RefAM mechanism.
>
> > **W3. Justification for Using DiT: While the paper highlights the application of Diffusion Transformer (DiT) as a key design choice, the rationale for this choice is not fully convincing. More quantitative or qualitative analysis demonstrating why DiT is particularly suitable—or superior—would strengthen the argument.**
>
> Our analysis is based on observations of GAS tokens and the interplay between text and visual tokens within multimodal cross-attention, phenomena that emerge only in transformer-based architectures. Moreover, the goal of this work is to investigate how state-of-the-art diffusion models, pretrained on large-scale datasets, capture semantic information and token interactions across modalities. All recent and SOTA diffusion models are built upon DiT architectures, therefore, adopting a DiT-based design is a natural and essential choice for our study.

---

> > ### Author Response · Authors · 2025-11-22
> > **Official Comment by Authors (2/2)**
> >
> > > **W4. Background on Stop Words as Attention Magnets: The experiments clearly show that introducing stop words as attention magnets improves performance. However, the theoretical or empirical background supporting this design choice is not well discussed. Providing references or related studies that justify the role of stop words in guiding attention could make the paper more comprehensive and grounded.**
> >
> > We thank the reviewer for their comment and we expand the discussion on the role of stop words as attention magnets below.
> > In our work, we empirically observe that stop words (including punctuation) act as attention attractors or “garbage collectors,” stabilizing multimodal attention maps in training-free diffusion settings. This phenomenon is consistent with several recent findings in NLP.
> >
> > Xiao et al. (ICLR 2024) first identified the attention sink phenomenon, where the model consistently allocates disproportionate attention to semantically light tokens such as the BOS marker. They also noted that the same mechanism can emerge around other non-semantic placeholders. More recent analyses extend this view: Zhang et al. (NeurIPS 2025) explicitly highlight punctuation and other low-semantic tokens as prompt-dependent attention sinks, showing that these tokens serve a similar stabilizing role by absorbing surplus attention mass. Empirical studies such as When Attention Sink Emerges in Language Models (ICLR 2025) further confirm that sink positions are not limited to the first token but can appear across various “light” or structurally uninformative tokens.
> >
> > Grounded in these observations, our use of stop words as attention magnets builds on and aligns with  previous and concurrent works that semantically light tokens can function as important anchors in attention dynamics. By leveraging this mechanism explicitly, we extend the attention-sink concept beyond single-modality applications to a broader, multimodal context.
> > We will include the discussion in the main.
> >
> > *References*
> >
> >  [1] Xiao et al., Efficient Streaming Language Models with Attention Sinks, ICLR 2024.
> >
> >  [2] Zhang et al., Attention Sinks: A “Catch, Tag, Release” Mechanism for Embeddings, NeurIPS 2025.
> >
> >  [3] Lee et al., When Attention Sink Emerges in Language Models, ICLR 2025.

---

### Official Review · Reviewer_KG5J · 2025-11-01

**Soundness:** 3
**Presentation:** 3
**Contribution:** 3
**Rating:** 6
**Confidence:** 4

**Summary:**

RefAM aims to address referring object segmentation in a zero-shot, training-free manner. It utilizes large pre-trained diffusion transformer models (DiTs) that encode cross-attention maps between text and image tokens. The key idea is the introduction of "attention magnets", i.e., stop words or additional tokens, that help redirect or absorb unwanted attention in the cross-attention maps. This leads to a more precise grounding of the referred object.

**Strengths:**

1. The discovery that stop words act as attention magnets and "global attention sinks" in diffusion transformers provides valuable insights.
2. This approach utilizes pre-trained diffusion models combined with attention manipulation, allowing the referring segmentation task to be performed without any additional training. As a result, it is widely applicable without the need for specialized labeled data.
3. On benchmarks such as RefCOCO, RefCOCO+, RefCOCOg, and video referring segmentation (e.g., Ref-DAVIS 17), it outperforms previous training-free methods.

**Weaknesses:**

1. The paper mentions "using diffusion transformer models," but it's crucial to note that the specific details such as the model variant, pretraining dataset, version, and hyperparameters are significant. Any variation in these aspects could influence the results. Since RefAM employs more robust vision backbones and pretrained models, it is difficult to attribute the improvements solely to the methodology rather than to these stronger backbones.
2. There is a reliance on large pretrained models, which necessitates powerful diffusion transformer backbones (DiTs) and segmentation models like SAM. This may lead to constraints in computational power or memory.
3. The quality of the masks may vary. The method extracts heatmaps and utilizes a segmentation model instead of being trained end-to-end for precise masks. As a result, the masks may be less accurate in complex scenes featuring occlusions or ambiguous expressions.
4. The stop-word and attention magnet strategy could be heuristic in nature. This strategy involves adding tokens and filtering them, which might require fine-tuning or may not generalize effectively across all languages or types of expressions.

**Questions:**

1. Which variants of the diffusion transformer (DiT) and FLUX were utilized in the RefAM experiments?
2. Can the same backbones and pretrained model be used for baseline comparisons?
3. How does RefAM perform when employing weaker or older vision-language backbones? Are performance gains still noticeable when adjusting for backbone strength?
4. How does RefAM compare to other zero-shot methods in terms of latency and memory usage?
5. Is the same set of magnets effective across different backbones and benchmarks?

---

> ### Author Response · Authors · 2025-11-22
> **Official Comment by Authors (1/3)**
>
> We thank the reviewer for the time taken to provide detailed and constructive feedback to our work. We appreciate that the reviewer acknowledges that our work "*provides valuable insights*" and "*is widely applicable without the need for specialized labeled data*".
>
> In the following, we would like to address the reviewer's remaining concerns.
>
> > **W1. … Since RefAM employs more robust vision backbones and pretrained models, it is difficult to attribute the improvements solely to the methodology rather than to these stronger backbones.
> Q2. Can the same backbones and pretrained model be used for baseline comparisons?**
>
> As shown in Table 6, removing the RefAM component (rows without the AM tick) leads to a substantial drop in performance, confirming that the observed improvements originate from our proposed method rather than from stronger backbones. Comparing the last row (our baseline) in Table 6 with the first row (HybridGL’s baseline) in Table 5 of the HybridGL paper further reveals that HybridGL’s baseline itself performs better than ours. We also note that the NP and SB components are directly adopted from the HybridGL method. Therefore, we attribute the significant performance gains primarily to the proposed RefAM mechanism.
>
> We would like to note that all previous methods rely on the CLIP model for zero-shot referral segmentation. In the following, we clarify why CLIP models are not suitable for our method. CLIP models are trained by aligning the EOS text token (summarizes the entire text input) with the visual CLS token (represents the overall visual input). Importantly, these models lack multimodal attention layers that connect individual text tokens with visual features. Such cross-modal attention is crucial for our method, as it enables fine-grained alignment between modalities and helps allocate redundancy to less informative tokens (e.g., stop words). In contrast, directly comparing each text token to global visual representations does not yield meaningful attention maps corresponding to distinct objects. Moreover, recent analyses show that CLIP token-level text features are weak representations of the sentence structure[1,2,3], which supports our observations. Therefore, our approach is not applicable to CLIP-style dual-encoder architectures.
>
> Additionally, we refer the reviewer to Table 11 in the appendix, where we evaluate our method on simple single-class segmentation tasks, such as ImageNet segmentation and VOC segmentation, and show improved performance against other methods using the same FLUX architecture.
>
> *References*
>
> [1] Yuksekgonul, Mert, et al. "When and why vision-language models behave like bags-of-words, and what to do about it?." ICLR 2023
>
> [2] Koishigarina, Darina, et al. "CLIP Behaves like a Bag-of-Words Model Cross-modally but not Uni-modally." arXiv preprint arXiv:2502.03566 (2025).
>
> [3] Tran, Allie, and Luca Rossetto. "On the Brittleness of CLIP Text Encoders." arXiv preprint arXiv:2511.04247 (2025).
>
> > **Q1. Which variants of the diffusion transformer (DiT) and FLUX were utilized in the RefAM experiments?**
>
> In our work, for image experiments, we use FLUX.1-dev. For video experiments, we utilize Mochi-1. We will add the details about model variants in the revised version. We would also like to note that our code will be released as open source to ensure transparency and reproducibility.

---

> > ### Author Response · Authors · 2025-11-22
> > **Official Comment by Authors (2/3)**
> >
> > > **W2. There is a reliance on large pretrained models, which necessitates powerful diffusion transformer backbones (DiTs) and segmentation models like SAM. This may lead to constraints in computational power or memory. Q4. How does RefAM compare to other zero-shot methods in terms of latency and memory usage?**
> >
> > We note that the SAM model is shared across all training-free referral segmentation methods. For SAM, approximately 3 GB of memory is sufficient to process a single image, which takes about 60 ms.
> > For FLUX-dev and Mochi, the requirements are higher, around 22 GB of memory and approximately 400ms per image, which is still reasonably fast processing. Our goal is to explore a new way of modeling semantic understanding in images using the current state-of-the-art diffusion models, without relying on the CLIP model from 2021 (CLIP ViT-B/16 requires about 2 GB and ~5ms/per image).
> >
> > Additionally, other approaches, e.g. HybridGL or Global-Local, rely on multiple inference passes to detect and combine various global and local representations. They also pre-extract segmentation proposals using a proposal network (SAM or FreeSolo), which substantially increases their computational pipeline and makes direct comparison of backbone efficiency less straightforward. E.g.  HybridGL’s total inference time, including this pre-extraction step, is ~1.1 seconds per image, whereas our method requires only ~460ms per image. Moreover, our method introduces a simpler and more direct pipeline for identifying relevant semantic information in images and videos, without additional auxiliary modules or pre-extraction steps.
> >
> > > **W3. The quality of the masks may vary. The method extracts heatmaps and utilizes a segmentation model instead of being trained end-to-end for precise masks. As a result, the masks may be less accurate in complex scenes featuring occlusions or ambiguous expressions.**
> >
> > We agree with an insightful observation of the reviewer. The quality of SAM masks is indeed a limitation. As discussed in Section D (Limitations) of our appendix, we acknowledge this limitation and include corresponding visualizations. However, we would like to emphasize that our method is training-free and relies on existing pretrained models that were not specifically designed for this task.
> >
> > Our motivation for using diffusion models lies in their pretraining on large-scale text–image datasets, which enables them to generate images that are semantically aligned with textual descriptions and capture complex semantic relationships. We believe that as diffusion models continue to improve, their ability to understand and extract complex scenes, including those with occlusions and ambiguous expressions, will improve accordingly.
> >
> > In contrast, previous methods rely on CLIP (a model from 2021), which is known to have limitations in semantic understanding [1] due to their global representations. With our approach, we propose a simple, training-free framework that can be readily adapted to current and also future diffusion models as they evolve.
> >
> > *References*
> >
> > [1] Tong, Shengbang, et al. “Eyes Wide Shut? Exploring the Visual Shortcomings of Multimodal LLMs.” Proceedings of the IEEE/CVF Conference on Computer Vision and Pattern Recognition, 2024.
> >
> > > **W4. The stop-word and attention magnet strategy could be heuristic in nature. This strategy involves adding tokens and filtering them, which might require fine-tuning or may not generalize effectively across all languages or types of expressions.**
> >
> > We thank the reviewer for this interesting observation. Our work specifically targets the training-free setting, and we build on our observation that stop words naturally act as “garbage collectors” in this context. We observe similar behavior across several diffusion models discussed earlier.
> >
> > In the NLP literature, it has been shown that the BOS token often functions as an attention sink. Furthermore, other non-semantic tokens can also attract a surplus of attention [1]. A recent study [2] further highlights prompt-dependent sinks, such as punctuation tokens, in addition to BOS. Therefore, we believe that our findings are consistent with ongoing research on attention sinks in NLP, vision, and multimodal models. Nonetheless, we agree that exploring a broader range of languages and expressions would be a valuable direction for future work.
> >
> > *References*
> >
> > [1] Xiao et al., “Efficient Streaming Language Models with Attention Sinks,” ICLR 2024.
> >
> > [2] Zhang et al., “Attention Sinks: A 'Catch, Tag, Release' Mechanism for Embeddings,” NeurIPS 2025.

---

> > > ### Author Response · Authors · 2025-11-22
> > > **Official Comment by Authors (3/3)**
> > >
> > > > **Q3. How does RefAM perform when employing weaker or older vision-language backbones? Are performance gains still noticeable when adjusting for backbone strength?**
> > >
> > > We adopt Stable Diffusion 3.5 (SD3.5) for our RefAM method. SD3.5 employs three text encoders simultaneously, two CLIP text encoders and one T5 text encoder, whose representations are concatenated. In contrast, FLUX uses two text encoders (T5 and CLIP), but only T5 is utilized for multimodal cross-attention conditioning, while Mochi includes only a single T5 text encoder.
> > >
> > > To enable fine-grained analysis of interaction between text and visual tokens in SD3.5, we separate the CLIP and T5 token streams and process them independently.
> > > The results on the RefCOCO testB split are presented in the table below.
> > >
> > > | Metric | Text Encoder | w/ SW | w/o SW |
> > > |:---|:---|:---:|:---:|
> > > | **oIoU** | T5 | 34.3 | 25.9 |
> > > | | CLIP | 27.0 | 30.5 |
> > > | | T5+CLIP | 28.4 | 30.2 |
> > > | **mIoU** | T5 | 41.6 | 31.7 |
> > > | | CLIP | 35.1 | 35.9 |
> > > | | T5+CLIP | 36.7 | 35.9 |
> > >
> > >
> > > We observe that using the T5 encoder yields significantly better performance than using either CLIP alone or the combined setup. These findings confirm that the CLIP encoder is not suitable for fine-grained text–visual alignment, supporting our earlier discussion (W1 and Q2).
> > >
> > > Moreover, we find a notable performance drop in the T5 model when stop words (SW) are removed, showing that our method generalizes across diffusion architectures. Interestingly, the CLIP encoder shows an opposite behavior, further reinforcing our observations regarding its different representation space. Similarly, combining T5 and CLIP representations leads to worse performance than using T5 alone, indicating that CLIP features may introduce confusion rather than complementarity.
> > >
> > > Finally, we note that, to the best of our knowledge, no current diffusion models integrate recent large language models (LLMs) as text encoders, which would allow for a deeper exploration of this phenomenon.
> > > We will update the main paper with the discussion and the results on all the datasets and subsets.
> > >
> > > > **Q5. Is the same set of magnets effective across different backbones and benchmarks?**
> > >
> > > We use the same set of magnets across backbones and benchmarks presented in the paper and for SD3.5.

---

> > > > ### Author Response · Authors · 2025-11-25
> > > > **Update: Full Evaluation on RefCOCO/+/g confirming backbone analysis for SD3.5**
> > > >
> > > > Following our previous response, we have now completed the experiments across the full RefCOCO, RefCOCO+, and RefCOCOg datasets. The results below strongly reinforce our initial observation regarding the architectural differences between T5 and CLIP within the SD3.5 framework.
> > > >
> > > > Key Observations:
> > > > 1. T5 is Structure-Aware: Across all datasets, the T5 encoder suffers a significant performance drop when stop words (SW) are removed (e.g., -10.9% mIoU on RefCOCO Test A). This confirms that T5 (and by extension our RefAM method) actively utilizes the syntactic structure of the prompt for grounding.
> > > > 2. CLIP acts as a "Bag-of-Words": The CLIP encoder is largely insensitive to syntax. Notably, on RefCOCO Test B, removing stop words actually improves CLIP's performance (+2.38% oIoU). This confirms that CLIP features prioritize object presence over spatial/relational structure, making them less suitable for precise referring expression generation.
> > > > 3. Redundancy of Combined Features: While SD3.5 defaults to concatenating T5 and CLIP, we find that T5 alone is often sufficient. In splits requiring strong spatial reasoning (RefCOCO), T5 matches or outperforms the combined model. Even in RefCOCO+ (where CLIP's appearance-focus is beneficial), the combined model yields only marginal gains over the individual encoders, suggesting that T5 provides the necessary structural backbone.
> > > >
> > > > These full-dataset results validate our design choice to rely on the T5 encoder for fine-grained text-visual alignment.
> > > >
> > > > | Dataset  | Split | Metric | T5 (w/ SW) | T5 (w/o SW) | CLIP (w/ SW) | CLIP (w/o SW) | T5+CLIP (w/ SW) | T5+CLIP (w/o SW) |
> > > > | :------- | :---- | :----- | :--------- | :---------- | :----------- | :------------ | :-------------- | :--------------- |
> > > > | RefCOCO  | Val   | oIoU   | 35.2      | 26.5       | 32.5        | 32.2         | 34.5           | 31.9            |
> > > > |   |    | mIoU   | 42.9      | 32.0       | 40.4         | 38.1         | 42.8           | 38.0            |
> > > > |   | TestA | oIoU   | 38.4      | 27.2       | 37.1        | 35.5         | 38.5           | 36.0            |
> > > > |   |  | mIoU   | 45.8      | 33.3       | 44.2        | 41.1         | 46.7           | 41.6            |
> > > > |   | TestB | oIoU   | 34.3 | 25.9       | 27.0 | 30.5         | 28.4           | 30.2            |
> > > > |   |  | mIoU   | 41.6      | 31.7       | 35.1        | 35.9         | 36.7           | 35.9            |
> > > > | RefCOCO+ | Val   | oIoU   | 27.8      | 22.3       | 28.6        | 25.2         | 28.7            | 24.9            |
> > > > |  |    | mIoU   | 34.1      | 26.2       | 34.8        | 29.7         | 35.5           | 29.7            |
> > > > |  | TestA | oIoU   | 30.2       | 23.2       | 31.4        | 27.5         | 31.8           | 27.8            |
> > > > |  |  | mIoU   | 37.5      | 28.1       | 38.3        | 33.1          | 39.5           | 32.9            |
> > > > |  | TestB | oIoU   | 25.0      | 21.0       | 24.2        | 23.1         | 24.2           | 21.7            |
> > > > |  |  | mIoU   | 29.8      | 24.5       | 29.4        | 27.1         | 29.4           | 25.2            |
> > > > | RefCOCOg | Val   | oIoU   | 27.6      | 25.7        | 26.6        | 25.8         | 27.0           | 24.5             |
> > > > |  |   | mIoU   | 35.0      | 31.9       | 34.0        | 31.4         | 34.3            | 30.4            |
> > > > |  | Test  | oIoU   | 28.0      | 27.2       | 27.81        | 26.3         | 28.0              | 25.9             |
> > > > |  |  | mIoU   | 34.9       | 32.8       | 34.63        | 31.9         | 35.0              | 31.8            |
> > > >
> > > > *w/SW corresponds to ReFAM

---

### Author Response · Authors · 2025-12-03
**Summary for Area Chairs**

Dear Area Chair,

We would like to provide a concise summary of the reviewers' positive assessments and how we have resolved the remaining inquiries.
# 1. Summary of Initial Reviews
The paper received positive initial ratings (6, 6, 4) with consensus on the method's effectiveness and clarity.
- **R-KG5J (6):** Commended the "valuable insights" regarding attention magnets and the method's "wide applicability" without training.
- **R-a3NV (6):** Highlighted the "state-of-the-art performance" and the "in-depth analysis" of global attention sinks.
- **R-xotR (4):** Stated the paper as "clearly written" and "well-motivated," acknowledging the SOTA results.
# 2. Summary of Discussion & Resolutions
We successfully addressed all concerns raised by the reviewers.
- **R-KG5J:**
  - *Summary of the main concerns:* The concerns focused on whether RefAM’s improvements are from the method itself or from the use of stronger backbones. Additional questions were regarding reliance on large pretrained models, and nature of the stop words. Clarification was also requested on performance with weaker backbones and latency, and the generalizability of the magnet set across models.
  - *Resolution:*
    - To further strengthen and support our claims, we provided ablations with a weaker backbone, in particular SD3.5 (Sec. 4.3). Results showed that removing our RefAM component causes a significant performance drop, proving the gains are method-specific.
    - We provided discussion regarding usage of pretrained large models including SAM (Sec. D). We also provided a latency comparison showing our method is faster (\~460ms) than competitors like HybridGL (\~1.1s) (Sec. 4.1).
    - We provided background on the role of semantically meaningless words in the NLP literature (e.g.Xiao et al.) 2024) that validates the existence of “attention sinks” in foundation models (Sec. 3.5).
    - Lastly, we confirmed to use the same set of stop words across FLUX, Mochi and SD3.5. Our code will be publicly available.
  - *Outcome:* Reviewer did not engage in the discussion.
- **R-a3NV:**
  - *Summary of the main concerns:* The concerns focused on generalization to different architectures such as variants of DiTs and CLIP models. The method’s reliance on SAM raised questions about fairness of comparison. Justification for selecting DiT architecture was asked. Additionally, despite effectiveness, RefAM design choice was asked to be further discussed or supported with relevant references.
  - *Resolution:*
    - We demonstrated generalization by extending our evaluation to SD3.5. Our method generalizes well across different DiT versions (FLUX, Mochi, SD3.5).
    - Following the reviewer’s recommendation, we investigated application of our method to the CLIP backbone. We note that SD3.5 includes two types of text encoders, CLIP and T5. We investigated the role of each encoder and observed more notable benefits for LLM-type backbone (T5) rather than CLIP, which is pretrained with contrastive learning. Our observation aligns with other literature on CLIP backbone, e.g. (Yuksekgonul, et al. 2023) that shows CLIP representations act as a bag of words structure.
    - Regarding SAM usage, we followed common practice in the IROS and VROS literature, therefore ensuring fair comparison with the previous literature.
    - We chose DiT architecture as we investigate cross-modal attention layers which induce emergence of the attention sinks, which justifies selection of this architecture.
    - To support our findings, we provided background on the role of semantically meaningless words in the NLP literature (e.g. Xiao et al. 2024) that validates the existence of "attention sinks" in foundation models (Sec. 3.5).
  - *Outcome:* The reviewer a3NV acknowledged that their concerns were addressed. However, shortly after the leakage their comment was deleted.
- **R-xotR:**
  - *Summary of the main concerns:* The concerns focused on novelty compared to "Vision Transformers Need Registers" (Darcet et al.).
  - *Resolution:* We clarified fundamental differences: our work **first** discovers **multimodal** attention sinks (vision-language), is **training-free** (uses existing stop words rather than training new tokens), and functions as a **filter** for noise rather than an accumulator of information.
  - *Outcome:* Reviewer did not engage in the discussion.
# 3. Concluding Remark
Thanks to the discussion, we have strengthened the manuscript by showing generalization to another DiT architecture, discovering distinct roles of T5 and CLIP text encoders, providing additional discussion regarding efficiency analysis and relation to recent NLP research. Given the strong initial recognition of our contributions (novel insights and SOTA results) and the comprehensive resolution of all technical queries, we believe RefAM offers a valuable contribution to the community. We are grateful for the AC's time and consideration.

Kind regards,

Authors of Submission 4001

---

### Meta-Review · Area_Chair_bA5q · 2026-01-06

**Summary:**

This submission was reviewed by three expert reviewers, with the ratings of: 2 borderline accept, and 1 borderline reject. The main concerns from the reviewers are around the strong backbones used, computations/memory cost due to the large models, quality of the masks, heuristic stop-word and attention magnet strategy, generalization issue, dependency on large models, justification of the technical designs, technical novelty and contributions. The authors provided a rebuttal for the concerns, but no further response was presented by the reviewers, nor discussion during the rebuttal phase.

After careful consideration by going through the review comments and the authors' rebuttal, it can be seen that some of the concerns raised by the reviewers have been addressed by the authors' rebuttal and experiments. However, most of the major concerns remain not well addressed (e.g. keep emphasising the training-free rather than addressing the corresponding concerns clearly). Overall, major concerns are not convincingly addressed, and there is no strong support for a clear acceptance. As a result, it is unfortunate that this submission in its current form is not ready to be accepted to ICLR.

**Reviewer Concerns:**

Concerns that the AC thinks were addressed by the rebuttal: the RefAM's performance when exploring weaker/older VL backbones; the novelty wrt prior works as pointed out by the reviewer is partially addressed, though not significant.

Concerns that are still outstanding: the backbone concern, though explained not convincing; the reliance on large pretrained models; the masks issues; concerns about the stop-word and attention magnet strategy; the generalization issue and the suggestions from the reviewer; justification for DiT; the significance of the technical novelty and contributions are still limited; efficiency concern and the reviewer's suggestion.

**Reviewer Scores:**

According to the review comments, and the rebuttal, for each review the reviewer might have changed their score in the way below, if they had been able to participate fully in the discussion:
* Reviewer KG5J: borderline accept to borderline reject, or reject
* Reviewer a3NV: borderline accept to borderline reject, or reject
* Reviewer xotR: borderline reject to reject.

---

### Decision · Program_Chairs · 2026-01-26

Reject